

# Development of a participatory Bayesian network model for integrating ecosystem services into catchment-scale water resources management

Jie Xue[1,2,3,4], Dongwei Gui[1,2], Jiaqiang Lei[1,2], Fanjiang Zeng[1,2], Rong Huang[1,2], Donglei Mao[5]

[1]State Key Laboratory of Desert and Oasis Ecology, Xinjiang Institute of Ecology and Geography, Chinese Academy of Sciences, Urumqi 830011, Xinjiang, China

[2]Cele National Station of Observation and Research for Desert-Grassland Ecosystems, Cele 848300, Xinjiang, China

[3]Key Laboratory of Biogeography and Bioresource in Arid Zone, Chinese Academy of Sciences, Urumqi 830011, Xinjiang, China

[4]University of Chinese Academy of Sciences, Beijing 100049, China

[5]College of Geographical Science and Tourism, Xinjiang Normal University, Urumqi 830054, Xinjiang, China

*Corresponding to*: Dongwei Gui (guidwei@ms.xjb.ac.cn), Jiaqiang Lei (desert@ms.xjb.ac.cn)

**Abstract.** This paper proposes an ecosystem services–based integrated water resource management (IWRM) framework within which a participatory Bayesian network (BN) model that assists with the integration of IWRM is developed. The framework is divided three steps: (1) identifying water-related services of ecosystems; (2) analysis of the tradeoffs and synergy among users of water; and (3) ecosystem services–based IWRM implementation using the BN model. We present the development, evaluation and application of a participatory BN model with the involvement of four participant groups (stakeholders, water manager, water management experts, and research team) in Qira oasis area, Northwest China. As a typical catchment-scale region, the Qira oasis area is facing severe water competition between the demands of human activities and natural ecosystems. We demonstrate that the BN model developed provides effective integration of ecosystem services into a quantitative IWMR framework via public negotiation and feedback. The network results, sensitivity evaluation, and management scenarios are broadly accepted by the participant groups. The intervention scenarios from the model conclude that any water management measure remains unable to sustain the ecosystem health in water-related ecosystem services. Greater cooperation among the stakeholders is highly necessary for dealing with such water conflicts. In particular, a proportion of the agricultural water saved through improving water-use efficiency should be transferred to natural ecosystems via water trade. The BN model developed is appropriate for areas throughout the world in which there is intense competition for water between human activities and ecosystems – particularly in arid regions.

## 1 Introduction

Water resource is a finite, vulnerable and also a scarce resource, essential for sustaining life, the environment and human development on the earth (UNEP, 2012; Bakker, 2012). Over few decades, with the increasing pressure from the growing human population, together with the spatiotemporal heterogeneity of the distribution of water resources against the background of climate variability, the provision of a reliable and available source of freshwater for human activities and ecosystem



demands has become a thorny issue worldwide (Bromley, 2005; Liu et al., 2013; Pang et al., 2014). Water competition caused by limited water supply for satisfying various requirements is the origin of many conflicts (Poff et al., 2003). Thus, there is an urgent need to develop principles and approaches to manage water efficiently, whilst at the same time respecting the requirements of

ecosystems (Cain, 2001; Bakker, 2012). The principles and approaches used to address this need will invariably involve a combination of biophysical, ecological, environmental, economic, social, cultural and political issues, as well as complex decision-related problems. Driven by these issues, an integrated water resource management (IWRM) plan has been proposed to promote coordinated development and water resources management via integrated assessment (Global

Water Partnership, 2000; Kragt, 2010; Siew and Doll, 2012). IWRM is internationally accepted as a good scheme for achieving sustainable development in a comprehensive and holistic manner (UNEP, 2012).

Two elements are essential in the various manifestations of IWRM. The first is that IWRM must be multidisciplinary in its approach. This implies the establishment of a particular framework in

which the evaluation of water management and decision-making cannot be restricted to the water resources alone; it must also incorporate the wide range of other factors into the IWRM framework (Bromley, 2005; Pollino and Henderson, 2010). The goal of multidisciplinary integration in this respect is to achieve synergy and tradeoffs between human demands and the maintenance of ecosystem health for freshwater (Bakker, 2012). Managing water between the two

ultimately seeks benefits obtainable from water allocation to maximize human wellbeing provided by ecosystem services, which are defined as a wide range of goods and services provided by ecosystems for human welfare (Millennium Ecosystem Assessment, 2005). According to Cook and Spray (2012), IWRM and ecosystem services have evolved into closely similar concepts, and face challenges linked to the coupling between them in terms of conceptualization and

implementation. In fact, the main problem at this juncture is that IWRM does not consider ecosystems as "users" of water in allocation (Jewitt, 2002). Therefore, an ecosystem services–based IWRM framework is highly necessary for building a bridge between the two concepts and for achieving sustainable water resource management.

The second claim is that management and decisions under IWRM must involve the participation

of stakeholders, as well as scientists and decision-makers, based on decision support system tools. The successful implement of an IWRM plan relies on the support of water-use departments in management strategies. Stakeholder involvement will provide effective coordination among various conflicts in the decision-making process, transparently and practically (Cain, 2001; Bromley, 2005; Kragt, 2009; Zorrilla et al., 2010). Moreover, the establishment and

implementation of an equitable and sustainable management scheme associated with stakeholders is quite complicated, and needs an efficient tool to complement the decisions. Decision support systems are suitable for providing a decision structure and to support "what-if" analysis of



possible decision-making options by using experimental data, model output, or expert knowledge (Cain, 2001; Chan et al., 2010). While a great many of the multidisciplinary approaches available – influence diagrams, decision trees, multi-criteria decision analysis, and so on – are broadly applicable, their employment is harder when it comes to dealing with specific environmental

modelling problems because of the complexity and uncertainty involved, and the added complication of stakeholder involvement in the IWRM (Bromley, 2005; Pollino and Henderson, 2010; Liu et al., 2013). Recently, considerable attention has been paid to Bayesian networks (BNs), which are graphical decision support system tools allowing "what-if" analysis through probability inference (Poppenborg and Koellner, 2014). BNs have been widely accepted as a popular

approach for modelling complex and uncertain issues associated with stakeholder participation (Uusitalo, 2007; Henriksen et al., 2007; Duspohl et al., 2012). Although stakeholder engagement in the decision process exchanges viewpoints to share new knowledge and solutions to common issues, few attempts have been made to confirm whether BNs developed by active stakeholder involvement and negotiation can assist and achieve common consensus to integrate ecosystem

services into IWRM.

The aim of this study is to establish an ecosystem services–based IWRM framework within which a participatory BN model could be developed for supporting sustainable IWRM. The framework presented in this paper can be divided into: identifying water-related services of ecosystems; analysis of the tradeoffs and synergy among users of water; and ecosystem services–

based IWRM implementation using the BN model. The participatory BN model is developed for application in the Qira oasis areas of Northwest China, to evaluate the framework associated with stakeholders. The case study area is broadly representative of many typical river catchments in which natural ecosystems face threats due to increasing water competition for drinking, domestic demands, industrial use, and agricultural irrigation. This paper is designed as the following

structure, the ecosystem services–based IWRM framework is firstly presented according to the characteristics of water use in the case study area. Next, the stakeholder representatives and participatory processes used to develop the conceptual BN model are described. And finally, the appropriateness of the BN model is evaluated and discussed through the sensitivity analysis, implementation of scenario simulations, and management decision recommendations.

## 2 Methods

### 2.1 Study area and IWRM issue

The Qira oasis area, also termed the Qira river oasis, is located in the lower reaches of the Qira River catchment of Northwest China (36°54′N–37°09′N, 80°37′E-80°59′E) and covers

approximately 274.63 km$^2$ (Figure 1). In comparison with most other river catchments, the Qira oasis area is a typical inland river catchment, situated between mountainous areas and amongst desert plains in an arid region. It is characterized by extremely low precipitation (39 mm/year),



strong evaporation (2700 mm/year), and highly vulnerable ecosystems (Bruelheide et al., 2003). The water supply in the Qira oasis area relies mainly on river discharge, which originates from a high altitude valley of the Kunlun Mountains, flowing through the Qira oasis area, and finally discharging into the extremely arid Taklimakan desert. According to monitoring data during 1960–

2010 from Qira hydrological station, the runoff in Qira River declined at a rate of $-0.003 \times 10^8$ m$^3$/year. Furthermore, with dramatic farmland extension in the upper reaches of the Qira River catchment, the extraction of river water for agricultural irrigation has led to the frequent drying-up of Qira River in the Qira oasis area (Xue et al. 2015). Numerous ecosystems in the Qira oasis area are experiencing serious issues due to the over-utilization of water for agricultural irrigation,

together with increasing domestic and industrial water demands.

The Qira oasis can be divided into agricultural and natural oasis ecosystems (Xue et al., 2016a; Xue et al., 2016b). The agricultural oasis ecosystem is essential for food production and human welfare, while the natural oasis ecosystem provides crucial ecosystem services for human survival

and settlement, such as desert vegetation acting as a natural barrier to maintain biodiversity, to protect agriculture, and to combat desertification and sandstorms. Aside from the other water use accounting for approximately 2.3% of the total water consumption, the other 97.7% is used to supply irrigation for agriculture – the main user of water. About 82.1% of the agricultural irrigation water is diverted from the Qira River, with the remaining 17.9% extracted from the oasis

area's groundwater (Hotan Water Resources Planning, 2013). To enhance the industrial proportion in economic structure, Xinjiang government in China encourages the investors to accelerate industrial development to lift more people out of poverty. The situation is increasingly aggravated by a lack of trade-offs and synergies between agricultural and natural oasis ecosystems together with intensive industrial water need.

The excessive diversion of water for agricultural irrigation threatens the health of natural oasis ecosystems. In Qira oasis area, the natural oasis ecosystem has been facing pressure due to water shortage. Many serious issues have emerged, including the destruction of the aquatic environment, the degradation of riparian forests and desert shrub-/grasslands, the deterioration of groundwater quality, and the decline of the groundwater table for maintaining the health of desert vegetation

(Xue et al., 2016b). Conversely, retaining a large amount of water for use by the natural oasis ecosystem can lead to a reduction in agricultural irrigation. It is not easy for agricultural irrigators, especially farmers, to agree to cut down on irrigation in order to cater for the water demands of the natural environment. Although Qira Water Conservancy Bureau is responsible for managing and allocating water to each water-use department, the coordination between agriculture and the

natural oasis ecosystem is very difficult in terms of meeting the needs and demands of the different stakeholders involved. Qira Agricultural Bureau is unwilling to reduce the level of irrigation so as to support the health of natural ecosystems, even though the water withdrawn from irrigation can provide potential benefits in return from the ecosystems. Qira Environmental




Protection Bureau claims that natural ecosystems, as the natural barrier for preventing desertification, should be given higher priority than other users of water. However, there are no tradeoff principles or approaches in place to deal with such conflicts associated with the various stakeholders in this region.

Managing water resources, based on the principles of IWRM together a decision support system tool, is of importance to achieve sustainable water development in the Qira oasis area. Since 2013, to ensure water resource security, Xinjiang's government has proclaimed "three red lines of water resource utilization" – water quantity, water quality, and water-use efficiency (Hotan Water Resources Planning, 2013). This water policy poses a considerable challenge in terms of

identifying reasonable water allocation and management strategies in a coordinated way in the Qira oasis area. In general, Qira oasis area was selected in this study because of its suitability as a universal representative of catchment-scale water management issues worldwide. Additionally, an important consideration is that the study area can easily obtain the available data under the support of Cele national station of observation and research for desert-grassland ecosystems, Chinese

Academy of Sciences.

## 2.2 BNs as decision support system tools for IWRM

BNs are probabilistic graphical models that conceptually represent a system as networks of interactions between variables via a cause–effect relationship diagram (Carmona, et al., 2011;

Chen et al., 2012). The probabilistic inference is implemented based on Bayes's paradigm. As a decision support system tool, a BN consists of two main components (Ropero et al., 2014): (1) a directed acyclic diagram (DAG), which is presented as a qualitative component and illustrated by directed arrows linking a set of variables or nodes with cause–effect relations; and (2) conditional probability tables (CPTs), regarded as a quantitative component. A variable or node comprises a

finite set of exclusive states that describe the "values" of variable discretization. The CPTs denote the strengths of the links expressed by conditional probability in the DAG. Figure 2 illustrates a simple example of a BN model. Figure 2a indicates a DAG with three variables: "Sprinkler", "Rain", and "The grassland is wet". Figure 2b shows the CPTs, consisting of a Boolean state ("Yes", "No") in each variable. For example, in the CPTs, the first value in the first column means

that when "Sprinkler" is "no" and "Rain" is also "no", then there is a 90% chance that "The grassland is wet" will be "no" (Xue et al., 2016b).

There is huge potential for the application of BNs in natural resources management, including IWRM (Kragt et al., 2011). BNs are widely considered suitable for integrating various issues and

investigating tradeoffs to model environmental systems (Chen and Pollino, 2012). Moreover, BNs can be readily built and understood by non-professional users and stakeholders due to their transparent graphical structure. This valuable characteristic of BNs can be developed into an





effective decision support system tool to support IWRM from transdisciplinary and participatory processes (Siew and doll, 2012; Mamitimin et al., 2015).

A detailed description of the Bayesian paradigm and probability propagation procedure can be found in the work of Fenton and Neil (2013). To be an effective decision support system tool, stakeholder involvement must play a crucial role in the DAG construction and analysis of BNs, especially the elicitation of the CPTs. Without stakeholder consultation, it is unlikely that a successful BN can be developed to implement IWRM and decisions (Cain, 2001). Furthermore, the compilation and implementation of a BN is dependent on the availability of associated software packages, including Hugin Expert (Hugin, www.hugin.com), Netica (Norsys Software Corp, www.norsys.com), AgenaRisk (AgenaRisk Software Package, www.agenarisk.com), and Analytica (Lumina Decision Systems, www.umina.com). In the present study, due to its flexibility and user-friendly interface, the popular Netica software package (Norsys Software Corp, http://www.norsys.com) is used to construct the network diagrams and to complete the inference.

## 3 Ecosystem services–based IWRM framework

The ecosystem services–based IWRM framework developed in this work is presented according to its three main steps: (1) identifying water-related services of ecosystems; (2) analysis of the tradeoffs and synergy among users of water; and (3) ecosystem services–based IWRM implementation using the BN model.

### 3.1 Identifying water-related services of ecosystems

The dependence of human wellbeing on services provided by ecosystems has been widely accepted by the general public (Millennium Ecosystem Assessment, 2005; Egoh et al., 2007; Egoh et al., 2008). Accordingly, ecosystem services cannot be substituted by other materials and technology, essential for human welfare and survival, directly and indirectly (Jewitt, 2002; Brauman et al., 2007; Nelson et al., 2009; Power, 2010). More importantly, ecosystem sustainability requires stable water supplies for use in water-related services of ecosystems to protect ecosystem functions (Jewitt, 2002). For comprehensive IWRM, achieving sustainable water resources management should consider ecosystems as one of the major users of water, to maintain ecosystem services and functions and thus ensure ecosystem health and sustainability. Ecosystems as users of water are becoming increasingly competitive with other users. To manage the quantity and quality of water in ecosystems, the identification of water-related services of ecosystems is indispensable for coordinating the balance of water between requirements and supply.

The ecosystems or sub-ecosystems, ecosystem service functions, users of water in suppliers of ecosystem services, and ecosystem disservices caused by water shortage, are identified and shown



in Figure 3 for the Qira oasis area. Since the Qira oasis area consists of agricultural and natural oasis ecosystems, the ecosystems were divided into seven sub-ecosystems associated with agriculture and the natural oasis environment. According to the characteristics of ecosystem services and functions, these sub-ecosystems could be classified into the corresponding functions of provisioning, regulating, supporting, and cultural service functions. Note that the ecosystem services and functions in the classification only represent the primary services and functions. For example, the agricultural ecosystem is both a provider and consumer of ecosystem services. Human beings value the agroecosystem chiefly for its provisioning services, such as food production, foraging and fiber supply. However, the agroecosystem also contributes cultural services (e.g., as an aesthetic landscape or generation of crop diversity) to the human population, often with spiritual comfort (Tallis et al., 2008).

Based on the characteristics of water utilization and consumption, users of water can be split into non-consumptive and consumptive users within the various ecosystems (Hong and Alexer; 2007; Savenije and Zaag, 2008). The non-consumptive users are often termed as the in-stream users of water, including the minimum river discharge for maintaining river ecosystem health, and groundwater restoration for ensuring groundwater system security. On the contrary, the remaining users are considered as consumptive users of water to embed the "virtual" water in the "products". For instance, the water for crops is consumed and embedded within agricultural products, expressed as typical consumptive users of water to guarantee food security. However, if the users of water do not have access to sufficient water supplies, undesirable disservices on ecosystems will emerge through water shortages. Therefore, with the intense competition for the limited freshwater resources in the Qira oasis area, the tradeoffs and synergy among users of water poses a considerable challenge when seeking to achieve sustainable IWRM.

### 3.2 Analysis of the tradeoffs and synergy among users of water

The relationship between ecosystem services and human wellbeing is described in the Millennium Ecosystem Assessment (2005). As an essential component, water supports the biosphere to enable the generation of ecosystem goods and services. Agricultural and natural ecosystems comprise the main ecosystem types in the Qira oasis area. While a large amount of water allocation for agroecosystems can increase provisioning ecosystem services, other supporting, regulating and cultural services provided by natural ecosystems, including sub-ecosystems, often suffer from losses and disservices due to water scarcity (Tallis et al., 2008).

The arbitrary supply of water for use in an ecosystem leads to disservices in other ecosystems (Tallis et al., 2008). The management of water in ecosystems has become vital for protecting ecosystem health and ensuring the sustainable use of ecosystem services. Integrated and coordinated assessment among multiple ecosystems is considered as an effective way to deal with





the water conflict among the users of water for water-related services of ecosystems. On the one hand, excessive water supply to support agricultural ecosystems can cause losses in natural sub-ecosystems, resulting in a win–lose scenario (Figure 4a). Likewise, a large amount of water to safeguard sub-ecosystem health can lead to disservices in the agroecosystem, also leading to a

win–lose scenario (Figure 4c).

On the other hand, sustainable integrated water management dealing with tradeoffs and finding synergy between agroecosystems and natural ecosystems can ultimately reach a win–win scenario (Figure 4b). Therefore, focusing on how users of water of ecosystems are integrated into IWRM is becoming an urgent need for achieving the sustainable use of ecosystem services and water

resources management.

### 3.3 Ecosystem services–based IWRM implementation using the BN model

IWRM is becoming an increasingly burdensome task that has to account for the interests of

multiple ecosystems. The sustainable use of ecosystem services needs to ensure non-consumptive and consumptive water supplies in ecosystems. Integrating ecosystem services into the IWRM framework must reduce tradeoffs and find synergy among the users of water for the for hydrologic ecosystems services.

Due to the capability of multidisciplinary modelling, BN models, as flexible and transparent

tools, have been widely used in ecosystem service modelling and water management (Carmona et al., 2011; Aguilera et al., 2011; Landuyt et al., 2013; Poppenborg and Koellner, 2014). In the present study, a participatory BN model was developed to implement the IWRM framework in which users of water for water-related services of ecosystems are embedded. Figure 5 illustrates the general layout of the BN for ecosystem services embedded in the IWRM framework. This

graphical representation shows the design of the structure of the BN model, which comprises three steps: analyze the available water supplies in the Qira oasis area; integrate users of water in the hydrologic ecosystem services in the network; and evaluate the benefit or disservice variables caused by water shortages. Moreover, the variable types in the structure are distributed in the corresponding framework.

### 4 Participatory BN model development

Public participation is becoming increasingly crucial in IWRM (Zorrilla et al., 2010; Liu et al., 2013). Active involvement and negotiation in the participatory process can effectively foster a

personal perspective for management strategies and the decision-making process, flexibly and transparently (Lynam et al., 2006, Reed, 2008; Wang et al., 2009; Carmona et al., 2011).





Furthermore, public participation is an essential part of the IWRM concept (Global Water Partnership, 2000), making the solution to a problem more straightforward and improving mutual understanding among water managers, domain experts and stakeholders (Mamitimin et al., 2009). Many studies have highlighted the importance of participation in system modelling and decision-making, especially in the IWRM setting (Henriksen et al., 2007; Chan et al., 2010; Zorrilla et al., 2010; Carmona et al., 2011). While a diverse set of participatory modelling tools exist that can be applied to implement the participatory process, BNs provide a potentially more effective alternative to achieve the goal of the decision support system because of the robustness of Bayes' theory and the visual nature of the software, which facilitates interaction and public participation (Cain, 2001).

Participatory BN models, or participatory BN modelling, are a specific subset of participatory modelling tools. The development of a BN model under public participation has been widely used in system modelling, and ultimately achieves a visual explanation of reality via the identification of key variables and their relationships (Lynam et al., 2007). Since variables in environmental system modelling are often difficult to quantify, usually due a poor understanding or lack of experimental data, the development of a participatory BN model is an essential task to support sustainable IWRM through participatory negotiation and evaluation (Zorrilla et al., 2010).

The development of a participatory BN model can be categorized into four phases: identification (identifying the problem and relevant variables), design (constructing the cause–effect diagram), implementation (BN inference), and evaluation (evaluating the model results) (Henriksen et al., 2007) (Figure 6). This comprehensive modelling process should be a recursive process, and ultimately obtains acceptable results from evaluation among stakeholders, water managers and domain experts. In addition, every phase also undergoes a recursive process in the public participatory process. This means that the discussion and negotiation among stakeholders is a spiral development process. Such a repetitive process will improve understanding and help to reach a consensus via public participation.

## 4.1 Public participatory process in BN model development

Public understanding of the environmental system can help to provide an integrated and qualitative representation of the catchment system, as well as for quantitative modelling (Chan et al., 2010). However, public perspective takes a long time to achieve and carries a large cost, despite detailed documentation available in the relevant literature and as part of local studies and reports (Cain, 2001). Public consultation and data collection are two major activities in the participatory process. According to Cain (2001) and Bromley (2005), public participants should include policy-makers and water management professionals, as well as the stakeholders in the IWRM. The involvement of decision-makers and experts can lead to a more comprehensive and





rigorous development of a system structure and management strategy. Extensive and detailed guidelines regarding the participatory process can be found in Cain (2001), Bromley (2005), Marcot et al. (2006), Kragt et al. (2009), and Pollino and Henderson (2010).

In Qira oasis area, the participants were divided into four participant groups: stakeholders, the water manager, water management experts, and researchers (Table 1). From a practical perspective, the number of stakeholders should be kept as small as possible, and also able to completely represent their own viewpoints (Burguess and Chilvers, 2006). Six departments were selected as the stakeholders in the Qira oasis area. Every department of stakeholders adopted two representatives (i.e., a head and professional of the department). To avoid conflict in discussion among stakeholders, the stakeholder meetings were implemented by the respective department with the research team. The meetings involved discussing water management problems from a general perspective. The policy-makers or decision-makers in the Qira oasis area were the water manager at the Qira Water Management Institute, which develops water policies and management plans. The face-to-face discussions led by the water manager focused on the management plans and strategies in the implementation of water policies. As a sub-group of the participants, six water management scientists from the research institute were involved in the expert knowledge consultation, as well as data elicitation and collation. In addition, the researchers were indispensable participants, serving various roles in the participatory process. The research team offered the participants a water management background and collected their feedback. More importantly, the team carried out the participatory procedure in a fair way via a two-way communication process (Rowe and Frewer, 2004, Charnley and Engelbert, 2005; Zorrilla et al., 2010).

The development of the BN model under public participation was organized into four procedures, beginning in March 2015 and ending in August 2016. Table 2 provides detailed information in this regard, including the objectives, meeting dates, organization format, participant groups, number of participants, and knowledge resources. While the participatory BN model development process involved four procedures, each step underwent a recursive or overlapping process during the participatory process. In general, the research team began by identifying the potential participants and by defining the issues. All the participants then determined the relevant variables and their relationships, whilst also designing the logic of the BN and eliciting reliable data from multiple resources. After the BN model was constructed, the researchers inserted the CPTs to analyze the results of the BN simulation. Finally, the developed BN model was evaluated and updated by all the participants in the participatory process. The process of evaluation was crucial for reaching a consensus, for achieving resonance among the participants, and for generating realistic results.





### 4.2 Model construction and data collection

The causal diagrams were built for direct application as the structure of the BN in an iterative process. Many rounds of stakeholder meetings and water manager interviews discussed and identified the plausible variables, states and structure. The research team then adjusted the relevant
interaction diagrams to build various causal networks. After the initial formation of conceptual graphs, water management experts were consulted to add, delete or improve the variables and states included in the networks, and even to modify the cause–effect links from their perspective. Many variables considered important but missing were added into the network, while those that presented limited relations were deleted from the diagrams. A complete and plausible structure of
the BN was ultimately determined after reaching a consensus among all the participants. The finalized structure of the BN, as well as the detail of the variables and states, is shown in Figure 7. A total of 56 variables were finalized, and the number of links was reduced to 74. This structure reduced the complexity of the BN model from the network of 56 variables having 3080 potential links.

To apply the BN model, quantitative data were obtained from various sources including the literature, empirical data, model output, government documents, official statistics, and expert interviews, to populate the CPTs. Ideally, the CPTs should be readily determined from the available dataset by an efficient parameter-learning algorithm (e.g., the maximum likelihood algorithm and EM algorithm). The empirical data, such as temperature, precipitation and river
discharge, were collected from Qira meteorological station and Qira hydrological station, and then processed. Other data, such as desert groundwater restoration, were obtained from model output. In particular, socioeconomic data, such as agricultural irrigation area, agricultural total output, and domestic water use, were collected from the statistical yearbooks of Xinjiang Province (2002–2013), Hotan Water Resources Planning (2013), and the Qira water resources planning report
(2013), and then analyzed. These data could be inputted in the Netica software package using the parameter learning algorithm.

However, many variables, such as policy data, are unmeasurable or irreproducible in the network. Expert knowledge plays an important role to elicit the CPTs. The selected experts were quite acquainted with the background and specialized in water management associated with
ecology, the environment, agricultural economics, and water policy. Through face-to-face interviews and consultations, the CPTs were elicited by expert knowledge and judgment discreetly. The elicitation process complies with the suggestion described by Cain (2001). The most extreme combinations of states are firstly populated in the table, and then the intermediate combinations are elicited through discussion and individual perception. The CPTs were finally averaged, based
on all the experts, for use in the BN.

### 5 Results and discussion



### 5.1 BN simulation analysis

The results of the BN simulations are illustrated as probability distributions by graphical modelling. Figure 8 shows the participatory BN simulation results in the current scenario. The probability of 63.3% for "riparian forest" (in the state of "over 17"), 59.3% for "desert vegetation" (in the state of "over 10.5"), 60.1% for "desert groundwater restoration" (in the state of "over 19.1"), and 66% for "minimum flow for river health" (in the state of "over 1.6"), indicates a greater than 50% likelihood of water provided from the Qira oasis area ensuring water-user health in water-related services of ecosystems. According to the cause–effect relationships of the network, these high likelihoods are explained by the frequent flood events (variable "flood") supporting these users of water to keep their health sustainable. Flooding in the Qira River basin usually occurs twice yearly: a spring flood caused by vast glacial and snow melting (variable "glacier and snow melting") due to an abrupt increase in alpine temperature (variable "temperature"); and a summer flood resulting from summer rainstorms (variable "precipitation") in the high-elevation mountain area (Chen, 2014). The water provided by flooding not only maintains the health of desert vegetation, forests and desert groundwater restoration, but also encourages the growth of new shrubs and plants in the seasonal flooding period (Bruelheide et al., 2003; Xue et al., 2015; Rumbaur et al., 2015). In the current scenario, this situation keeps the modeling results basically consistent with precious evidence (Xue et al., 2015) and with stakeholder perspectives.

However, due to water shortages and competition in such an arid area, the probability for the other three   users of water – the urban greenbelt water (variable "water for urban greenbelt"), man-made shelterbelt water demand (variable "water for man-made shelterbelt"), and agricultural irrigation (variable "agricultural irrigation quantity") – is relatively low. It has been confirmed that the urban greenbelt provides important ecosystem services in detaining dust, as well as in beautifying the city (Kretinin and Selyanina, 2006; Liu et al., 2013). The urban greenbelt is the first defense against the sand and dust storms that are frequent in this region. Unfortunately, the water supply for the urban greenbelt is at present inadequate. The stakeholders, especially farmers, are more inclined to allocate vast quantities of water to agriculture under water shortage conditions. Moreover, such fervent competition for water in this limited water resource area has led to the water supply for the man-made shelterbelt and for agricultural irrigation to be insufficient. According to the Hotan Water Resources Plan (2013), the likelihood of achieving >2716.8 million $m^3$ and >100625.1 thousand $m^3$ water for the man-made shelterbelt is only 16.2% and 34%, respectively. Currently, such a situation only serves to increase the challenge in achieving coordination between water for agriculture and the environment.

The benefit or disservice variables caused by sufficient or insufficient water supplies for users of water in hydrologic ecosystem services are shown in the seven output variables. The probability for biodiversity (variable "biodiversity"), groundwater safety (variable "groundwater safety"),





drinking-water security (variable "8.6-25.6 thousand people"), grassland degradation (variable "grassland degradation") and agricultural income (variable "agricultural income") presents a "medium" likelihood or degree in the probability distribution. From the propagation structure of the network, such "medium" likelihoods are mainly attributed to "medium" users of water. For example, a "medium" degree of water supply for the man-made shelterbelt indicates a "medium" likelihood of grassland degradation under the "normal grazing" condition. In comparison with the above five variables, land desertification has been gradually improved (72.9% likelihood for less than 259.77 km$^2$ in land degradation area), implying that the man-made shelterbelt has to a significant extent prevented land degradation despite a "medium" likelihood in the water supply for the man-made shelterbelt. On the contrary, soil salinization remains serious (36.7% likelihood for 16.8–21 ha) owing to low water saving efficiency (<0.43) and poor salt-removing systems and devices.

### 5.2 BN model evaluation

The developed BN model needed to be evaluated after constructing the "cause–effect" relationship structure and eliciting the CPTs. The model assessment tools included qualitative evaluation (e.g., the participatory feedback from stakeholders and experts (Zorrilla et al., 2010)) and quantitative validation (such as the evaluation of predictive accuracy by comparison with observed data or results from other models (Poppenborg and Koellner, 2014), and sensitivity analysis (Kragt, 2009; Chan et al., 2010)). Of these two types of model evaluation tools, sensitivity analysis is widely regarded as the more effective method to assess model performance (Cain, 2001; Bromley et al., 2005; Marcot et al., 2006; Pollino and Henderson, 2010). Sensitivity analysis was therefore used in the present study to test the sensitivity of the BN outcome variables to variations in input parameters. Moreover, mutual information (see Pearl (1988) and Barton et al. (2008)) was considered as the measure of the sensitivity analysis to perform the BN model evaluation.

In general, the objective variables of the network were used to test which variables impacted on the target variables with high sensitivity (Chan et al., 2010; Poppenborg and Koellner, 2014; Xue et al., 2016). In this study, the seven benefit or disservice variables were set as the target variables to perform the sensitivity analysis. Figure 9 displays the results of the sensitivity analysis for the benefit or disservice variables. The left side of the vertical coordinate denotes the mutual information value, while the right side refers to the variance of beliefs. Visually, the length of the blue bars corresponding to each sensitivity variable in the figure is a measure of the influence of that variable on the target variable. The larger the mutual information value is, the more sensitive the influencing variable is on the target variable.

The influences of drinking-water engineering and groundwater quality on the variable "drinking-water security"; grazing and the man-made shelterbelt on the variable "grassland degradation"; crop yields, spring irrigation and irrigation quota on the variable "agricultural



income"; and salt-removing system and water-saving efficiency on "soil salinization" are all very sensitive. Interpreting the sensitivity of these variables is fairly straightforward. For instance, the augmentation of crop yields and adequate spring irrigation can increase agricultural income, verifying the reasonability that crop yields directly decide agricultural income, and spring

irrigation as the key period of crop water requirement indirectly impacting crop yields.

Furthermore, the river and natural oasis ecosystem, groundwater restoration and groundwater depth, as well as the man-made shelterbelt and desert vegetation, are more sensitive than the other variables in the analysis of the variables "biodiversity", "groundwater safety", and "land desertification", respectively. Since the impact of the other variables in the BN gradually decreases

as the number of intermediate variables increases (Marcot et al., 2006; Poppenborg and Koellner, 2014), These sensitivity results match well with anecdotal evidence and with stakeholder perspectives. Taking the variable "biodiversity", for example, the river's aquatic organisms and natural vegetation are essential for maintaining biodiversity, supporting the interpretation that the vulnerability of these two ecosystems, especially the former, impacts greatly on the biodiversity in

the Qira oasis area.

### 5.3 Scenario analysis and management

Having constructed and evaluated the BN model, it could be used to analyze the scenario

simulation of the relative likelihood of changes in target variables associated with variations in management actions. The impact of one or more input variables on the others could be easily predicted by specifying the state of those input variables. Table 3 lists the percentage changes in the probability of user variables of water for water-related services of ecosystems accompanying a specified state, given different intervention implementations. In comparison with the current

scenario, a groundwater extraction plan and the digging of wells are able to increase the likelihood of water supply for the urban greenbelt (37.6%), for the man-made shelterbelt (68.1%), and for agricultural irrigation (13.1%). It is clear that these two interventions indirectly expect to extract water from groundwater for users of water, increasing the likelihood of water supply. Furthermore, the building of reservoirs together with sufficient/insufficient funds and planning can lift the

possibility of agricultural income (21.6%). Obviously, because it accounts for 35% of agricultural water demand in spring, the building of reservoirs can store water to ensure spring irrigation, relieving extreme shortages of spring irrigation in the Qira area.

However, on the contrary, the intervention actions associated with the building of reservoirs decreases the likelihood of water demand for riparian forest (−13%), desert vegetation (−7.4%),

and desert groundwater restoration (−12.9%). This is because the reservoirs are built in the upper reaches of the Qira River basin, i.e., the headwater of Qira oasis. Once the river water is retained





and cannot reach the oasis area or lower reaches, the likelihoods of ensuring riparian forest, desert shrub-/grassland vegetation, and desert groundwater restoration, are inevitably decreased. In addition, the execution of the three red lines (−22.1% likelihood) and the water price (−14.2% likelihood) can reduce the quantity of agricultural irrigation for saving the proportion of

agricultural water.

Correspondingly, Table 4 shows the changes in benefit or disservice variables resulting from management actions. The intervention action associated with building reservoirs can decrease the likelihood of biodiversity (−5.5%), groundwater safety (−5.2%), and land desertification (−6%). However, this action is likely to increase agricultural income (4.3%) due to ensuring spring

irrigation. While the intervention associated with increasing water extraction from groundwater decreases the possibility of groundwater safety (−28.9%), it can instead improve the likelihood of grassland degradation (7.6%), land desertification (17.7%), and agricultural income (0.4%). In addition, the provision of advanced engineering and devices is necessary, because good drinking-water engineering and salt-removing systems are able to increase the likelihood of

drinking-water security (36.5%) and soil salinization (22.7%) quite considerably.

The overall results of the model's application can demonstrate plausible and useful management suggestions under different intervention scenarios to water managers and stakeholders. The model's outcomes imply that management in the form of integrating ecosystem services into IWRM needs greater cooperation from the stakeholders, as well as control from the

water managers. On the one hand, the stakeholders require a deeper understanding of ecosystem services, which can bring irreplaceable benefits and thus ensure the responsibility of water for hydrologic ecosystem services is shared among the stakeholders. For example, riparian forest and desert shrub-/grassland vegetation are the main "defense lines" in combating desertification and sandstorms, as well as for supporting biodiversity. Supplying water through flooding to ensure the

health of these aspects becomes crucial in the benefits of ecosystem services. On the other hand, socioeconomic demand is absolutely essential in providing sufficient water to boost agricultural development. Spring irrigation accounts for 35% of irrigation's annual total in the Qira oasis area, and its shortage is a continuously serious issue, leading to significant reductions in agricultural production. Therefore, building reservoirs to store river water in the upper reaches provides spring

irrigation and relieves the agricultural water shortage in spring. However, building reservoirs results in a shortage for ecological users of water. Regularly drawing off river water after building reservoirs is a plausible way to coordinate water conflict between agriculture and natural ecosystems. This process can be completed by water trade between stakeholders.

## 5.4 Challenges and prospects for participatory BN model development



While a participatory BN model can successfully be used to assist in solving the issue of integrating ecosystem services into IWRM, the participatory procedure of stakeholders such as farmers' representatives is a very time-, energy- and money-consuming process, in terms of introducing the research background and achieving a methodological understanding of BNs, as well as with respect to making appointments for meetings or interviews. Unlike other tools that can complete the system modelling process within a day, developing a full BN takes much time to implement, owing to the stakeholder and expert consultation, as well as the data collection and collation. On the one hand, encouraging interpretation of the research topic and providing accessibility to the BN process makes the stakeholders and experts more accepting and convinced; while on the other hand, the meetings require financial investment in terms of the time of the experts and expenses for arranging workshops. In particular, time with experts often has to be postponed and rearranged due to the water management experts' other commitments.

Active public involvement and negotiation help to build a transparent and flexible BN model by collecting and structuring stakeholder and expert knowledge. However, participant knowledge is often perceived as subjective information, which potentially can lead to biased outcomes (Uustialo, 2007; Pollino and Henderson, 2010). The experts' judgements tend to be prone to under- or overconfidence in terms of quantitative estimates, resulting in uncertainty when knowledge and data are limited (Uustialo, 2007). Moreover, frequent consultation can cause the participants to be reluctant or impatient, particularly in the elicitation of the CPTs.

In the present study, it took approximately one and a half years to develop the BN. In order to construct a plausible structure and elicit the relevant CPTs, the stakeholders and experts have to be familiar with the issue of IWRM associated with ecosystem services. A better understanding of the model will enhance the rationality of the BN model, avoiding and reducing the subjective bias provided by limited knowledge. The assessment process mainly depends on expert knowledge and literature results to validate the model. In general, this study effectively developed a BN model to integrate ecosystem services into IWRM through public participation. Our work expands the concept of IWRM by considering the importance of ecosystem services, thus helping to provide holistic water resources management through the participatory BN tool.

Although a participatory BN model is poor at representing the spatiotemporal characteristics of dynamic processes, and is limited in a number of other ways (as described above), the tradeoffs among stakeholders combined with expert knowledge can successfully offer assistance to decision-makers and water managers to deal with water-use conflicts with straightforward and easily understandable characteristics (Cain, 2001; Bromley, 2005; Pollino and Henderson, 2010). In comparison with other modelling approaches, a participatory BN model will provide the advantage of integrating different factors and options, such as ecosystem services, into system modelling through public discussions (Pollino and Henderson, 2010). More importantly, another



advantage of participatory BN models is the ease with which the existing model can be updated when new knowledge and data become available (Landuyt et al., 2013). Therefore, it is necessary to carry out further research that focuses on the integration of multiple perspectives, as well as ecosystem services, into IWRM projects, using participatory BN models. Also, the BN model

constructed in the present study should be updated in a timely manner as and when new knowledge and data become available, to improve the accuracy of its simulation.

**6 Conclusions**

Water resource management has undergone a major transition from multi-purpose management to

transdisciplinary integrated basin management. In particular, ecosystem services–based governance is increasingly being pushed in the direction of IWRM owing to a wide range of ecosystem services–related benefits to human wellbeing. In large part, IWRM is creating an opportunity to achieve the juncture or coupling with ecosystem services. To successfully achieve a coupling between ecosystem services and IWRM, the ecosystems need to be considered as users

of water alongside other users.

BNs represent an effective framework that can allow the integration of different knowledge into system modelling. More importantly, BN models are able to engage stakeholders in the management and decision-making process, dealing explicitly with the source of uncertainty in the participatory process. Public participation (e.g., the involvement of stakeholders and domain

experts) plays a crucial role in sharing system understanding and in strengthening the participants' sense of ownership and responsibility. In particular, due to the lack of quantitative data, the inclusion of dispersed knowledge is essential to develop a robust model, and to test the constructed model through participant discussions and negotiation in the IWRM.

This paper proposes an ecosystem services–based IWRM framework to develop a BN model

under public participation. The Qira oasis area, Northwest China was selected as a typical catchment-scale region to construct and verify the participatory BN model, since 97.7% of the water in this region is used for agricultural irrigation, leading to degradation of the natural ecosystem through intense water conflict. The model's structure and results were eventually accepted following many discussions and negotiations among participant groups as part of the

participatory process. Currently, no single water management scenario is able to sustain the ecosystem health in water-related services of ecosystems in the Qira oasis area. Greater cooperation from stakeholders is recommended for dealing with such water conflicts – in particular, by establishing a water trade mechanism and improving the water-use efficiency in agricultural irrigation, which saves some of the water to be used by users of water in natural

ecosystems.

The BN model developed in the present study confirms that a participatory BN is a feasible



tool for integrating ecosystem services governance into sustainable IWRM through social learning, thus effectively addressing the reality under limited available data. It also shows the potential for assisting in catchment-scale synergy and tradeoffs between agriculture and natural ecosystems. More importantly, the BN model provides an open and transparent system to support IWRM

decision-makers, such as water managers and environmentalists, to prioritize management interventions and to optimize the returns to expected objectives such as ecosystem services. However, the uncertainty in the participatory process caused by poor knowledge and understanding, as well as a lack of data, needs to be addressed in future research.

**Acknowledgements**

This work was financially supported by the National Natural Science Foundation of China (41601595, 41471031), the Task 2 of the Key Service Project 5 for the Characteristic Institute of CAS (TSS-2015-014-FW-5-3), and the Project of Science and Technology Service Network Initiative of CAS (KFJ-SW-STS-176). The authors would like to acknowledge all stakeholders

and experts who participated in the study. Moreover, we also wish to thank Professors Fengqing Jiang, Hailiang Xu, Zhiming Qi, Guojun Liu, Zhenyong Zhao, and Lei Wang for their enthusiastic support and valuable contribution in the interviews, and for their suggestions during review of the manuscript.

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





Table   Captions

Table 1. Stakeholder groups selected in the Qira oasis area.

Table 2. Participatory BN model development process in the Qira oasis area (March 2015 to August 2016).

Table 3. Influence of scenario management on water-user variables for water-related services of ecosystems in the BN simulation (the positive and negative values in the table denote the probability difference (%) between management and the current scenario).

Table 4. Impact of scenario management on benefit or disservice variables in the BN simulation (the positive and negative values in the table denote the probability difference (%) between management and the current scenario).





Table 1. Stakeholder groups selected in the Qira oasis area.

| Participant group | Department of participants | No. of participants | Position(s) in the participant groups |
|---|---|---|---|
| Stakeholders | Water Conservancy Bureau | 2 | Head of Qira Water Conservancy Bureau and professional |
| | Agricultural Bureau | 2 | Head of Qira Agricultural Bureau and professional |
| | Meteorological Bureau | 2 | Head of Qira Meteorological Bureau and professional |
| | Environmental Protection Bureau | 2 | Head of Qira Environmental Protection Bureau and professional |
| | Forestry Bureau | 2 | Head of Qira Forestry Bureau and professional |
| | Village committee | 2 | Village head and representative of farmers |
| Water manager | Water Management Institute | 1 | Head of Qira Water Management Institute |
| Water management experts | Xinjiang Institute of Ecology and Geography | 6 | Scientists of Xinjiang Institute of Ecology and Geography |
| Researchers | Research team | 9 | Professors, doctoral and masters students in research team |





Table 2. Participatory BN model development process in the Qira oasis area (March 2015 to August 2016).

| Participatory process | Objectives | Date | Format | Participants (no.) | Knowledge resource |
|---|---|---|---|---|---|
| Identification | 1) Identify potential participants 2) Identify the relevant variables 3) Identify the possible scenarios | March 2015 | Group meeting | Research team (9) | Literature review, professional knowledge |
| Design | 1) Construct the logic of the BN 2) Obtain the relevant data from multiple resources | September 2015 | Group meeting, stakeholder interview, expert interview, water manager interview | Research team (9), stakeholders (12), expert team (6), water manager (1) | Literature review, professional knowledge, expert knowledge |
| Implementation | 1) Insert the CPTs into the BN 2) Implement the BN model and analyze results | January 2016 | Group meeting | Research team (9) | Literature review, professional knowledge |
| Evaluation | 1) Evaluate the model results 2) Recommend the scenario management | August 2016 | Group meeting, stakeholder interview, expert interview, water manager interview | Research team (9), stakeholders (13), expert team (6), water manager (1) | Literature review, professional knowledge, expert knowledge |



Table 3. Influence of scenario management on water-user variables for hydrologic ecosystem services in the BN simulation (the positive and negative values in the table denote the probability difference (%) between management and the current scenario).

| Intervention variable | Water-user variables for hydrologic ecosystem services | | | | | | |
|---|---|---|---|---|---|---|---|
| | RF | MFRH | DV | DGR | WUG | WMS | AIQ |
| Building reservoirs | −6.9 | 0 | −3.9 | −6.8 | 0 | 0 | +20.9 |
| Digging wells | 0 | 0 | 0 | 0 | +16.4 | +30 | +5.8 |
| Groundwater extraction plan | 0 | 0 | 0 | 0 | +21.2 | +38.1 | +7.3 |
| Execution of three red lines | 0 | 0 | 0 | 0 | 0 | 0 | −22.1 |
| Execution of water price | 0 | 0 | 0 | 0 | 0 | 0 | −14.2 |
| Funds for building reservoirs | −4.2 | 0 | −2.4 | −4.2 | 0 | 0 | +0.5 |
| Building reservoirs plan | −1.9 | 0 | −1.1 | −1.9 | 0 | 0 | +0.2 |
| Subsidy for high-tech irrigation | 0 | 0 | 0 | 0 | 0 | 0 | 0 |
| Economic compensation policy | 0 | 0 | 0 | 0 | 0 | 0 | 0 |
| Water diversion project | 0 | 0 | 0 | 0 | 0 | 0 | 0 |
| Drinking-water engineering | 0 | 0 | 0 | 0 | 0 | 0 | 0 |
| Water-saving engineering | 0 | 0 | 0 | 0 | 0 | 0 | 0 |
| Grazing | 0 | 0 | 0 | 0 | 0 | 0 | 0 |
| Salt-removing system | 0 | 0 | 0 | 0 | 0 | 0 | 0 |

Note: RF, riparian forest (state of "over 17"); MFRH, minimum flow for river health (state of "over 1.6"); DV, desert vegetation (state of "over 10.5"); DGR, desert groundwater restoration (state of "over 19.1"); WUG, water for urban greenbelt (state of "over 80"); WMS, water for man-made shelterbelt (state of ">2716.8"); AIQ, agricultural irrigation quantity (state of ">100625.1").





Table 4. Impact of scenario management on benefit or disservice variables in the BN simulation (the positive and negative values in the table denote the probability difference (%) between management and the current scenario).

| Intervention variable | Benefit or disservice variables | | | | | | |
|---|---|---|---|---|---|---|---|
| | BI | GS | DS | GD | LD | AI | SS |
| Building reservoirs | −2.9 | −2.7 | 0 | 0 | −3.2 | +2.3 | 0 |
| Digging wells | 0 | −12.7 | 0 | +3.2 | +7.6 | +0.2 | 0 |
| Groundwater extraction plan | 0 | −16.2 | 0 | +4.4 | +10.1 | +0.2 | 0 |
| Execution of three red lines | 0 | 0 | 0 | 0 | 0 | -1.2 | 0 |
| Execution of water price | 0 | 0 | 0 | 0 | 0 | -1.2 | −1.4 |
| Funds of building reservoirs | −1.8 | −1.7 | 0 | 0 | −1.9 | +1.4 | 0 |
| Building reservoirs plan | −0.8 | −0.8 | 0 | 0 | −0.9 | +0.6 | 0 |
| Subsidy of high-tech irrigation | 0 | 0 | 0 | 0 | 0 | +1.4 | −1.8 |
| Economic compensation policy | 0 | 0 | 0 | 0 | 0 | +4.5 | −1.8 |
| Water diversion project | 0 | 0 | 0 | 0 | 0 | 0 | 0 |
| Drinking-water engineering | 0 | 0 | +36.5 | 0 | 0 | 0 | 0 |
| Water-saving engineering | 0 | 0 | 0 | 0 | 0 | +3.7 | −4.6 |
| Grazing | 0 | 0 | 0 | +17.9 | 0 | 0 | 0 |
| Salt-removing system | 0 | 0 | 0 | 0 | 0 | 0 | +22.7 |

Note: BI, biodiversity (the state of "good"); GS, groundwater safety (state of "high"); DS, drinking-water security (state of "<8.6"); GD, grassland degradation (state of "slight"); LD, land desertification (state of "<104.26"); AI, agricultural income (state of ">0.35"); SS, soil salinization (state of "<10.08").




Figure Captions

Figure 1. Location of the Qira oasis area.

Figure 2. An example illustrating the DAG and CPTs of a BN with three Boolean variables: (a) the DAG – the qualitative component of the BN; (b) the CPTs – the quantitative component of the BN.

Figure 3. Identification of water-related services of ecosystems.

Figure 4. "Scenario flowers" representing the tradeoffs and synergy between water for agricultural ecosystem and other ecosystems.

Figure 5. General layout of the BN for ecosystem services embedded in the IWRM framework.

Figure 6. Public participatory BN development processes as a recursive process to support ecosystem services–based IWRM (illustrated in the inner circle), and also discussion and negotiation from stakeholders and experts in every phase (shown in the outer circle).

Figure 7. Participatory BN model for ecosystem services–based IWRM developed by active involvement and negotiation of stakeholders and domain experts.

Figure 8. Participatory BN model simulation with elicited CPTs.

Figure 9. Sensitivity analysis for various benefit or disservice variables.


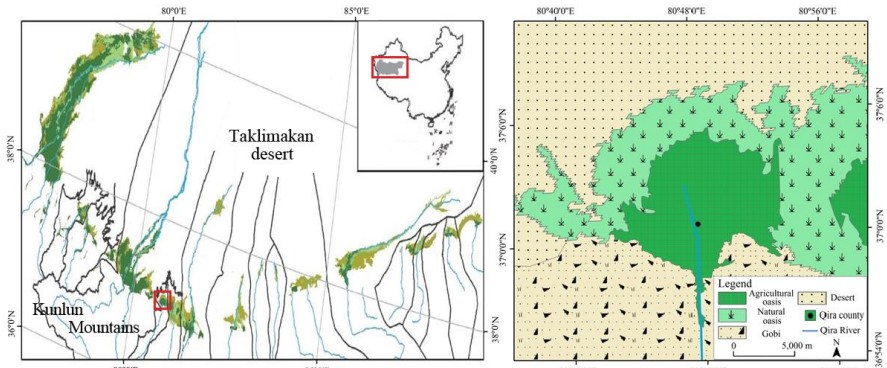

Figure 1. Location of the Qira oasis area.





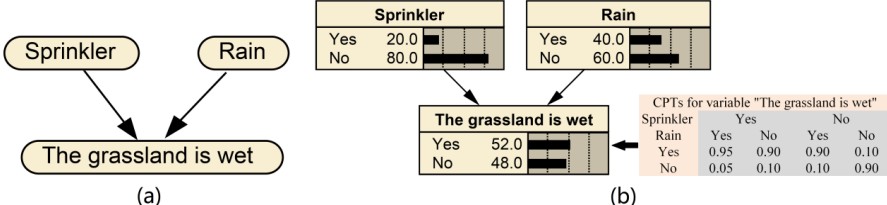

Figure 2. An example illustrating the DAG and CPTs of a BN with three Boolean variables: (a) the DAG – the qualitative component of the BN; (b) the CPTs – the quantitative component of the BN.




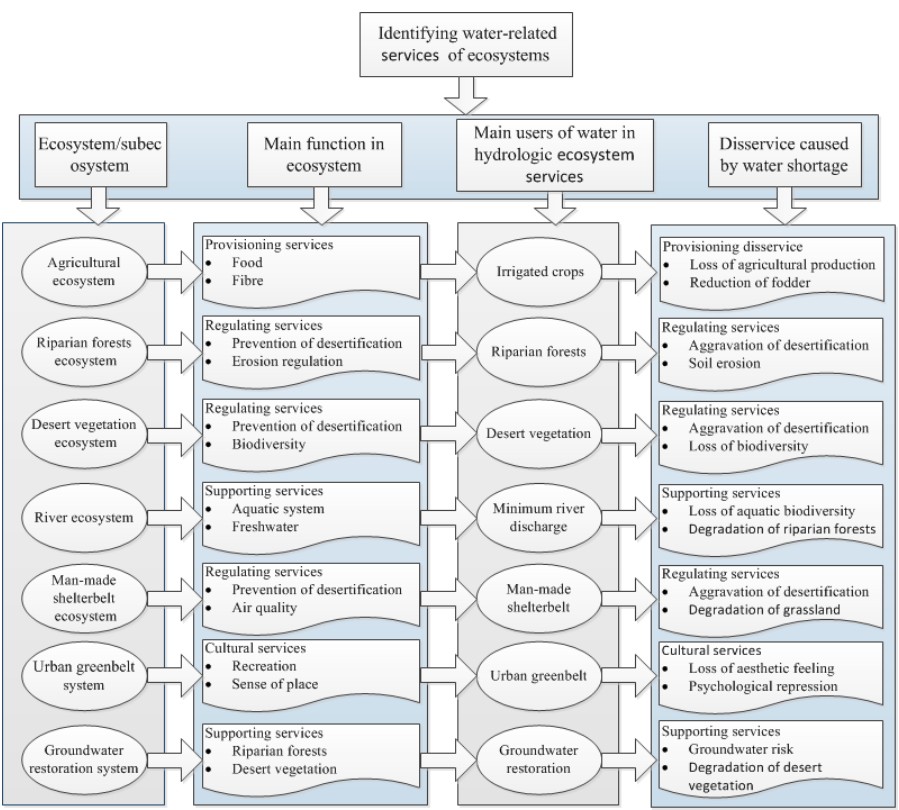

Figure 3. Identification of water-related services of ecosystems.




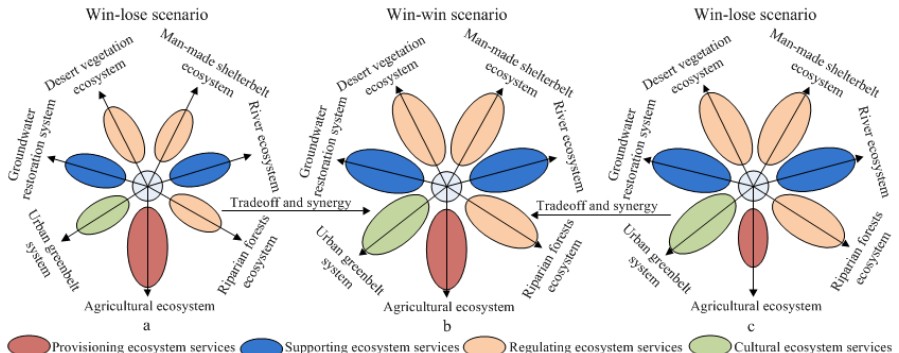

Figure 4. "Scenario flowers" representing the tradeoffs and synergy between water for agricultural ecosystem and other ecosystems.





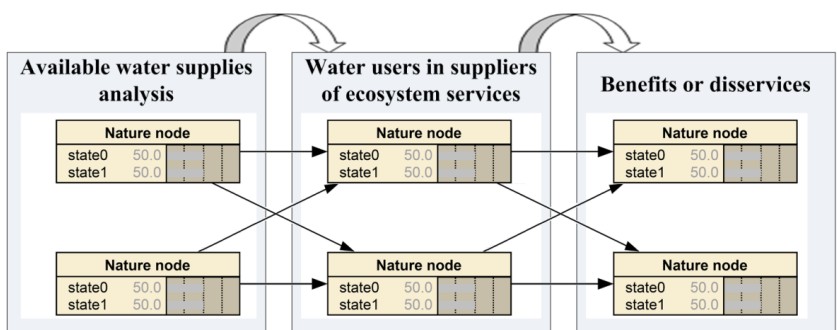

Figure 5. General layout of the BN for ecosystem services embedded in the IWRM framework.



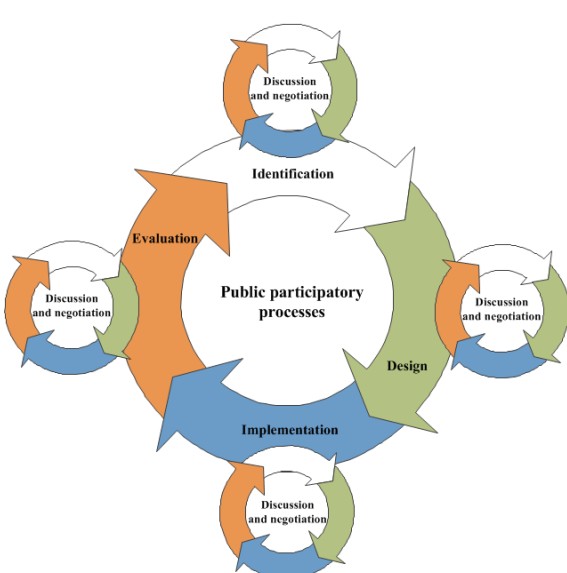

Figure 6. Public participatory BN development processes as a recursive process to support ecosystem services–based IWRM (illustrated in the inner circle), and also discussion and negotiation from stakeholders and experts in every phase (shown in the outer circle).



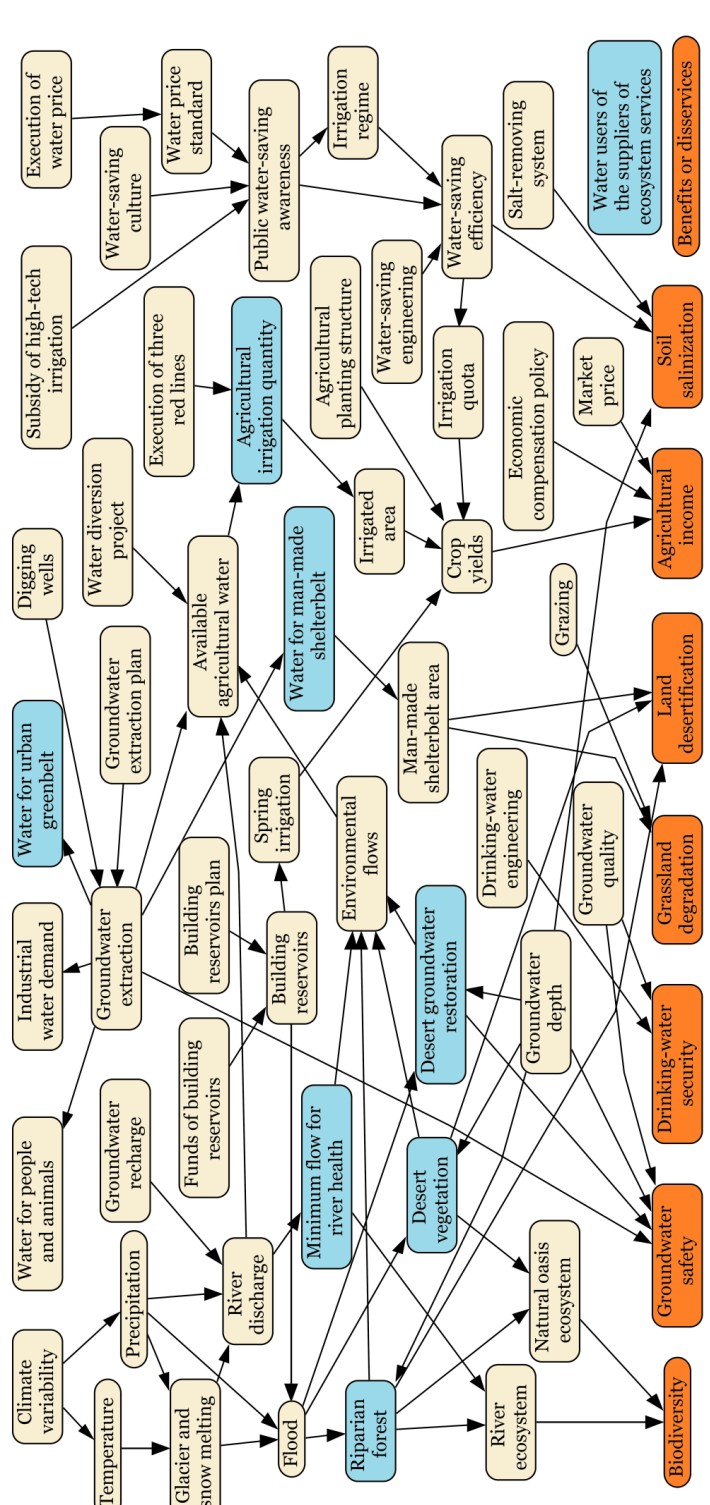

Figure 7. Participatory BN model for ecosystem services—based IWRM developed by active involvement and negotiation of stakeholders and domain experts.







Figure 8. Participatory BN model simulation with elicited CPTs.





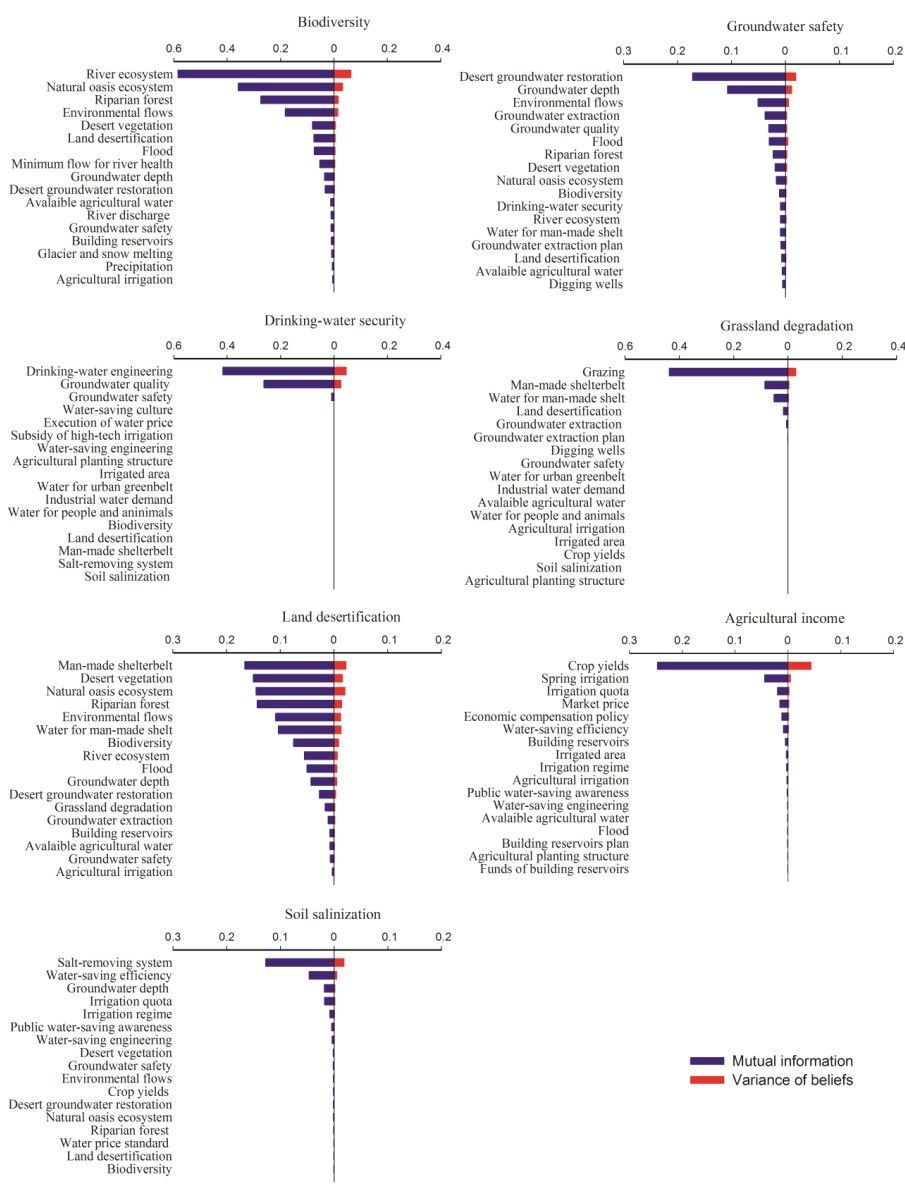

Figure 9. Sensitivity analysis for various benefit or disservice variables.