# Peer review of "Development of a participatory Bayesian network model for integrating ecosystem services into catchment-scale water resources management"

_Hydrology and Earth System Sciences, 2016_

## Referee Comment (RC1) · M. McClain (Referee) · 19 Feb 2017

This manuscript reports the outcome of an in-depth and extensive process of stakeholder engagement, translated into a Bayesian network model of a catchment-scale water resource management situation in the Qira oasis area of western China. The aim of the study was to establish an ecosystem services-based IWRM framework supported by the BN model. Components of the framework include identification of the water related services and the analysis of trade-offs between these and other services. Over an 18 month period the researchers met with stakeholders to identify key variables and design the model logic and then to evaluate the model results and scenarios for balancing water use.

The results of the model for the current scenario illustrate the limitations of the available resource to simultaneously meet water related ecosystem services as well as consumptive demands related to agriculture and local green infrastructure. The model was then used to evaluate other scenarios with stakeholders, illustrating the extent and type of trade-offs required when prioritising one or another water demand. No scenario was able to produce a situation acceptable to all stakeholders. The authors weigh the advantages and disadvantages of using BN models in such a participatory process, emphasising the time and cost associated with proper model development.

The framework is quite basic and applies BN models in a widely used manner. It is therefore not a novel framework but does represent an interesting application in the Qira area. The results and discussion topics are also common in such BN model applications. Another common feature of the study is a level of complexity and opaqueness in the model that makes it difficult for readers to unpack and understand the analysis and results.

My main recommendation for the authors is therefore to provide additional information (in a supplementary file) to better explain the various nodes and causal linkages. Supplementary data and explanations should also be provided for the CPTs.

Additionally, the methodology for applying the scenario management reported in Table 3 is not clear. Were the input intervention variables simply turned on or off and the effect noted? "Building reservoirs" is labelled an intervention variable but is itself dependent on other intervention variables included in the table. More detail on how the scenarios were analysed would present a clearer picture to readers.

I also offer the following observation to help improve the manuscript.

Pg 4 line 16: The statement that desert vegetation "protects" agriculture is provocative.

[Figure]

Additional explanation would be helpful here.

Pg 5 section 2.2: The first paragraph of this section it too basic. BNs are sufficiently established and known. I recommend simply noting the methodology used and not explaining the value and rationale of BNs.

Pg 6 line 22-23: I wish there was "wide acceptance" of the dependence of human well-being on ecosystem services, but I don't know what evidence supports this. There has certainly been a lot of academic attention, arguments made by NGOs, and language introduced into policy. I suggest changing this to "growing acceptance".

Pg 6 line 24: Being explicit about which services cannot be substituted would be helpful here, because of course many can.

Pg 8 line 33: Public participation has "always" been crucial to IWRM. It is one of the fundamental principles. This is then mentioned again on pg 9 line 1. Check manuscript for redundancies, which are common.

Pg 11 line 3: "many rounds" of stakeholder meetings are mentioned but I see evidence of only two rounds in Table 2. Is there another stakeholder process not represented in Table 2?

Pg 11 line 13: It is an exaggeration to cite 3080 potential links because they are only potential if there is a meaningful relationship. I suggest deleting this sentence.

Pg 11 line 25: "and then analysed" What analysis was conducted?

English is generally quite good but a careful review and correction of grammatical errors is needed. E.g. pg 5 line 5, pg 8 line 17... many more.

---

## Referee Comment (RC2) · Anonymous Referee #2 · 3 Mar 2017

Review comments to hess-2016-618: Development of a participatory Bayesian network model for integrating ecosystem services into catchment-scale water resources management submitted by Jie Xue et al.

This paper presents an ecosystem services-based IWRM framework to develop a participatory Bayesian network model for assisting integration of IWRM, dealing with an interesting and complicated issue. Authors described the development, evaluation, and application of a participatory Bayesian network (BN) model with four participant groups involved in the Qira oasis area, Northwest China. The case analysis in the Qira oasis

area demonstrated the availability of ES-based IWRM framework using the participatory Bayesian network model.

The BN developed effectively provided the integration of ES into quantitative IWMR framework via the public negotiation and feedback. The results reported that any water management measure from BN model still cannot sustain the health of suppliers of ES in the Qira oasis area. In general, the authors illustrated some merits in the methods and results. Furthermore, the authors also pointed out the contribution and challenge of their work in the Result and discussion.

This paper has a good potential to be published in the journal. However, there are some significant issues, listed below, which need to be addressed before it is ready for publication. 1. Please delete the redundant phrase "at the same time" in Page 2, Line 4.

2. The expression in Page 3, Lines 27-29 is not proper. It should be described by using two sentences.

3. The sentences in Page 6, Line 31 is confusing. Maybe that can be changed in "users of water within ecosystems are becoming increasingly competitive with other users"? Please clarify.

4. Please delete the redundant "for" in Page 8, Line 17.

5. In page 9, line 15: it should be "due to" rather than "due".

6. I think the Section 5.1 is exchanged with Section 5.1 in Results and discussion. After a model is developed, the first thing is to evaluate the reasonability of the model. It can then model simulation or prediction, and so forth.

7. Please double check the grammatical errors in the text. For example, the word "was" should be changed in "is" in Page 5, Line 11.

8. Please change "These" into "these" in page 14, Line 11.

9. The description in Page 15, Lines 3-5 is confusing. Please clarify.

10. The accuracy of some word uses should be double checked in the manuscript. For example, the word "stakeholder" should be replaced by "participant" in the Table 1 of Page 24.

Please also note the supplement to this comment:
http://www.hydrol-earth-syst-sci-discuss.net/hess-2016-618/hess-2016-618-RC2-supplement.pdf

---

## Author Comment (AC1) · 27 Mar 2017

**Response to Reviewer #2:**

**Title:** *Development of a participatory Bayesian network model for integrating ecosystem services into catchment-scale water resources management*
**Authors:** *Jie Xue, Dongwei Gui, Jiaqiang Lei, Fanjiang Zeng, Rong Huang, Donglei Mao*
**Manuscript ID:** *hess-2016-618*

**The authors would like to thank you for the time you invested in reviewing this manuscript. We would also like to thank you for your insightful comments on this revised version of the manuscript.**

**According to your comments, we have carefully modified the manuscript. The corresponding revised manuscript with the modifications shown in the document is attached following the responses to the comments.**

This paper presents an ecosystem services-based IWRM framework to develop a participatory Bayesian network model for assisting integration of IWRM, dealing with an interesting and complicated issue. Authors described the development, evaluation, and application of a participatory Bayesian network (BN) model with four participant groups involved in the Qira oasis area, Northwest China. The case analysis in the Qira oasis area demonstrated the availability of ES-based IWRM framework using the participatory Bayesian network model.
The BN developed effectively provided the integration of ES into quantitative IWMR framework via the public negotiation and feedback. The results reported that any water management measure from BN model still cannot sustain the health of suppliers of ES in the Qira oasis area. In general, the authors illustrated some merits in the methods and results. Furthermore, the authors also pointed out the contribution and challenge of their work in the Result and discussion.
This paper has a good potential to be published in the journal. However, there are some significant issues, listed below, which need to be addressed before it is ready for publication.

**Response: Thank you for your positive comments. Below are our responses to the pertinent issues. We hope that we have adequately addressed all your comments.**

1. Please delete the redundant phrase "at the same time" in Page 2, Line 4.
**Response: We have deleted it in the manuscript as suggested**.

2. The expression in Page 3, Lines 27-29 is not proper. It should be described by using two sentences.
**Response: We have improved that sentence as suggested. Please see the changed sentence in the manuscript.**

3. The sentences in Page 6, Line 31 is confusing. Maybe that can be changed in "users of water within ecosystems are becoming increasingly competitive with other users"? Please clarify.
**Response: Thank you for highlighting this mistake. We have changed it as "The users of water within ecosystems are becoming increasingly competitive with other users". Please see the changed sentence in the manuscript.**

4. Please delete the redundant "for" in Page 8, Line 17.
**Response: We have deleted it in the manuscript as suggested**.

5. In page 9, line 15: it should be "due to" rather than "due".
**Response: It has been changed in the manuscript as suggested**.

6. I think the Section 5.1 is exchanged with Section 5.1 in Results and discussion. After a model is

developed, the first thing is to evaluate the reasonability of the model. It can then model simulation or prediction, and so forth.

**Response: Thank you for your constructional suggestion. The Section 5.1 is exchanged with Section 5.2 in the Results and discussion. Please see the changed sentence in the manuscript.**

7. Please double check the grammatical errors in the text. For example, the word "was" should be changed in "is" in Page 5, Line 11.

**Response: Thank you for reminding us of the grammatical errors in the manuscript. To improve the grammatical errors of our manuscript, we have invited a professor in the field of water resources, who is a proficient English speaker, to go through our manuscript. Please see the changed part in the manuscript.**

8. Please change "These" into "these" in page 14, Line 11.

**Response: It has been changed in the text as suggested.**

9. The description in Page 15, Lines 3-5 is confusing. Please clarify.

**Response: Thank you for pointing this out. We have improved it in the text. Please see the changed sentence in the manuscript.**

10. The accuracy of some word uses should be double checked in the manuscript. For example, the word "stakeholder" should be replaced by "participant" in the Table 1 of Page 24.

**Response: Thanks for pointing out the incorrect word use. We have carefully checked the correctness of main word use in the text. Please see the changed words in the manuscript.**

[revised manuscript text omitted]

---

## Author Comment (AC2) · 27 Mar 2017

**Response to Reviewer #1:**

**Title:** *Development of a participatory Bayesian network model for integrating ecosystem services into catchment-scale water resources management*
**Authors:** *Jie Xue, Dongwei Gui, Jiaqiang Lei, Fanjiang Zeng, Rong Huang, Donglei Mao*
**Manuscript ID:** *hess-2016-618*

**The authors would like to thank you for the time you invested in reviewing this manuscript. We would also like to thank you for your insightful comments on this revised version of the manuscript.**

**According to your comments, we have carefully modified the manuscript. The corresponding revised manuscript with the modifications shown in the document is attached following the responses to the comments.**

This manuscript reports the outcome of an in-depth and extensive process of stakeholder engagement, translated into a Bayesian network model of a catchment-scale water resource management situation in the Qira oasis area of western China. The aim of the study was to establish an ecosystem services-based IWRM framework supported by the BN model. Components of the framework include identification of the water related services and the analysis of trade-offs between these and other services. Over an 18 month period the researchers met with stakeholders to identify key variables and design the model logic and then to evaluate the model results and scenarios for balancing water use.

The results of the model for the current scenario illustrate the limitations of the available resource to simultaneously meet water related ecosystem services as well as consumptive demands related to agriculture and local green infrastructure. The model was then used to evaluate other scenarios with stakeholders, illustrating the extent and type of trade-offs required when prioritising one or another water demand. No scenario was able to produce a situation acceptable to all stakeholders. The authors weigh the advantages and disadvantages of using BN models in such a participatory process, emphasising the time and cost associated with proper model development. The framework is quite basic and applies BN models in a widely used manner. It is therefore not a novel framework but does represent an interesting application in the Qira area. The results and discussion topics are also common in such BN model applications. Another common feature of the study is a level of complexity and opaqueness in the model that makes it difficult for readers to unpack and understand the analysis and results. My main recommendation for the authors is therefore to provide additional information (in a supplementary file) to better explain the various nodes and causal linkages. Supplementary data and explanations should also be provided for the CPTs.

**Response: Thank you for your constructive comments. We agree that the BN model developed is somewhat complicated, and lack of adequate explanation for readers. As you said, we have not provided additional information to detail the variables, states, and data information. To explain more clearly, we have added a supplementary file as Appendix A to provide detailed information about variables, states, and data information. This Appendix A provides a summary of the variables, states, and information sources used to elicit the CPTs. Moreover, the sentence "The states representing each variable and information used to elicit the CPTs are explained in Appendix A" is inserted in the manuscript. Please see the added Appendix A in the revised manuscript with the modifications following the responses to the comments.**

Additionally, the methodology for applying the scenario management reported in Table 3 is not clear. Were the input intervention variables simply turned on or off and the effect noted? "Building reservoirs" is labelled an intervention variable but is itself dependent on other intervention variables included in the table. More detail on how the scenarios were analysed would present a clearer picture to readers.

**Response: Thanks for your pertinent suggestion. The scenario managements described in Table 3 are the changes that are produced in the management objective variables when the values of each intervention variable are changed. As you said, an intervention variable may be relevant with other intervention variables, but the effects of combination among intervention variables can be obtained by the sum of all interventions (Cain, 2001). Therefore, we mainly analyzed the changes of management objective variables when each intervention variable is conducted as scenario management. We have improved the description in Table 3 in the text. Please see the changed sentences in the revised manuscript.**

**Cited references are listed as follows:**

**Cain J. 2001. Planning Improvements in Natural Resources Management. Guidelines for Using Bayesian Networks to Support the Planning and Management of Development Programmes in the Water Sector and Beyond, Centre for Ecology and Hydrology: Wallingford, UK, 2001. pp.89-94.**

I also offer the following observation to help improve the manuscript.

Pg 4 line 16: The statement that desert vegetation "protects" agriculture is provocative. Additional explanation would be helpful here.

**Response: Thank you for pointing this out. We wanted to express that the desert vegetation can prevent soil erosion and land degradation in the farmland to "protect" agriculture. To state more clearly, we have changed "to protect agriculture" in "to prevent farmland erosion and degradation".**

Pg 5 section 2.2: The first paragraph of this section it too basic. BNs are sufficiently established and known. I recommend simply noting the methodology used and not explaining the value and rationale of BNs.

**Response: To simplify the description of BNs, we have deleted Figure 2 and its explanation in the manuscript as suggested.**

Pg 6 line 22-23: I wish there was "wide acceptance" of the dependence of human wellbeing on ecosystem services, but I don't know what evidence supports this. There has certainly been a lot of academic attention, arguments made by NGOs, and language introduced into policy. I suggest changing this to "growing acceptance".

**Response: We agree. We have change "widely accepted" to "growingly accepted".**

Pg 6 line 24: Being explicit about which services cannot be substituted would be helpful here, because of course many can.

**Response: Thanks for your suggestion. We have improved it in the text. We agree that some ecosystem services can be substituted by other materials and technology, but the others cannot. For example, desert vegetation including desert forests and desert shrubs in arid regions plays an irreplaceable role in preventing soil erosion and desertification (Xue et al., 2015; Rumber et al., 2015). Although the protective engineering can overcome them, it is too limited. We have changed "ecosystem services" as "some ecosystem services such as desert vegetation preventing soil erosion and land degradation" Please see the changed sentences in the manuscript.**

**Cited references are listed as follows:**

**Xue J, Gui D W, Zhao Y, et al. 2015. Quantification of environmental flow requirements to support ecosystem services of oasis areas: a case study in Tarim Basin, Northwest China, Water, 7, 5657–5675.**

**Rumbaur C, Thevs N, Disse M, et al. 2015. Sustainable management of river oases along the Tarim River (SuMaRiO) in Northwest China under conditions of climate change, Earth Syst. Dynam., 6, 83–107.**

Pg 8 line 33: Public participation has "always" been crucial to IWRM. It is one of the fundamental principles. This is then mentioned again on pg 9 line 1. Check manuscript for redundancies, which are common.

**Response: We have deleted and improved some redundancies in the text. Please see the changed sentences in the manuscript.**

Pg 11 line 3: "many rounds" of stakeholder meetings are mentioned but I see evidence of only two rounds in Table 2. Is there another stakeholder process not represented in Table 2?

**Response: Thank you for pointing this out. In the development process of BN model, the participatory processes of stakeholders are repetitive processes. Namely, the design and evaluation phases of BN model underwent recursive processes in the participatory process of stakeholders. In Table 2, although the stakeholders mainly participated in the design and evaluation phases of BN model, the stakeholder meetings are many times with formal and informal forms. Therefore, to avoid such the misunderstanding, we have deleted "many rounds**

**of" in the text.**

Pg 11 line 13: It is an exaggeration to cite 3080 potential links because they are only potential if there is a meaningful relationship. I suggest deleting this sentence.

**Response: We have deleted that sentence as suggested in the text.**

Pg 11 line 25: "and then analysed" What analysis was conducted?

**Response: We wanted to explain that the obtained data was then analyzed. To express it more clearly, we have changed it as "and then the obtained data was analyzed".**

English is generally quite good but a careful review and correction of grammatical errors is needed. E.g. pg 5 line 5, pg 8 line 17… many more.

**Response: Thank you for highlighting the grammatical errors. To improve the grammatical errors of our manuscript, we have invited a professor in the field of water resources, who is a proficient English speaker, to go through our manuscript. Please see the changed part in the manuscript.**

[revised manuscript text omitted]

Appendix A. Variables, variable states, detailed explanation, and information sources used in eliciting CPTs.

| Variable | States | Explanation | Information sources |
|---|---|---|---|
| Climate variability | Yes, no | Climate change impacts on the variation of water resource | Literature values (Xue et al. 2015) |
| Water-saving culture | Good, poor | Water-saving awareness in Muslim religious culture | Survey results |
| Groundwater depth | <4, 4-10, >10 | Groundwater depth (m) | Qira water resources planning report (2013) |
| Groundwater quality | <1, 1-3, >3 | Groundwater quality (g/l) | Qira water resources planning report (2013) |
| Groundwater recharge | <22.63, 22.63-29.20, >29.20 | Groundwater recharge (million m$^3$) | Hotan Water Resources Planning (2013) |
| Building reservoirs | Yes, no | Building reservoirs to relieve the pressure among water demands | Results of stakeholder interviews |
| Digging wells | Yes, no | Exploiting groundwater based on groundwater resource evaluation | Results of stakeholder interviews |
| Groundwater extraction plan | Increasing, decreasing | Groundwater extraction policy | |
| Execution of three red lines | Good, poor | Water policy from quantity, quality, and water-using efficiency | Results of stakeholder interviews |
| Execution of water price | Good, poor | Water considered as good to increase water-saving consciousness | Results of stakeholder interviews |
| Funds of building reservoirs | Sufficient, insufficient | Support of fund is indispensable for building reservoirs | Results of stakeholder interviews |
| Building reservoirs plan | Yes, no | Building reservoirs policy | Results of stakeholder interviews |
| Subsidy of high-tech irrigation | High, low | Economic stimulation for promotion of high-tech irrigation | Results of stakeholder interviews |
| Economic compensation policy | Yes, no | Economic compensation policy in three red lines | Results of stakeholder interviews |
| Water diversion project | Yes, no | Water diversion plan for ensuring water supply | Results of stakeholder interviews |

| Drinking-water engineering | Good, poor | Engineering plan for ensuring drinking-water health | Results of stakeholder interviews |
| Water-saving engineering | Good, poor | Anti-seepage engineering of channels | Results of stakeholder interviews |
| Grazing | Overgrazing, normalgrazing | Grazing intensity in the human activities | Results of stakeholder interviews |
| Groundwater extraction | <22.80 ; 22.80-23.26 ; >23.26 | Groundwater extraction in water consumption (million $m^3$) | Qira water resources planning report (2013) |
| Irrigated area | <8057, 8057-11326, >11326 | Agricultural irrigated area (ha) | Statistical Yearbooks of Xinjiang Province (2002–2013) |
| Water price standard | <0.02, 0.02-0.05, >0.05 | Water price standard (RMB/$m^3$) | Results of stakeholder interviews |
| Man-made shelterbelt area | <1071, 1071-2240, 2240-3500, 3500-3850, >3850 | Man-made shelterbelt area (ha) | Results of stakeholder interviews |
| Irrigation quota | <8142, 8142-9857, 9857- 10728, 10728-12128, >12128 | Agricultural irrigation quota($m^3$/ha) | Qira water resources planning report (2013) |
| Agricultural planting structure | Plan 1, plan 2, plan3 | Cultivated area: forest area: pasture area= 61.22:36.49:2.29 (Plan 1), 50.36:47.39:2.25(plan 2), 43.60:54.45:1.95 (plan 3) | Hotan Water Resources Planning (2013) |
| Environmental flows | <40.29%, 40.29%-50.84% , 50.84%-53.48%, 53.48%-58.75%, >58.75% | Percent of river runoff | Calculated outputs in the model (Xue et al. 2015) |
| Temperature | <0.44, 0.44-1.37, >1.37 | Annual mean temperature (℃) | Literature values (Xue et al. 2015) |
| Precipitation | <134.48, 134.48-162.02, >162.02 | Annual accumulated precipitation (mm) | Literature values (Xue et al. 2015) |
| Glacier and snow melting | <51, 51-63, >63 | Annual glacier and snow melting (million $m^3$) | Hotan Water Resources Planning (2013) |
| Flood | Increasing, decreasing | Flood events | Results of stakeholder interviews |
| River discharge | <104 , 104-129, >129 | Annual river discharge (million $m^3$) | Literature values (Xue et al. 2015) |
| Riparian forest | Under 17, over 17 | Water demand for riparian forest (million $m^3$) | Calculated outputs in the model (Xue et al. 2015) |

| Minimum flow for river health | Under 1.6, over 1.6 | Minimum flow for ensuring river health (million $m^3$) | Calculated outputs in the model (Xue et al. 2015) |
|---|---|---|---|
| Desert vegetation | Under 10.5, over 10.5 | Water demand for desert vegetation (million $m^3$) | Calculated outputs in the model (Xue et al. 2015) |
| Desert groundwater restoration | Under 19.1, over 19.1 | Desert groundwater restoration (million $m^3$) | Calculated outputs in the model (Xue et al. 2015) |
| River ecosystem | <1.6, 1.6-5, >5 | Water demand for ensuring river ecosystem (million $m^3$) | Calculated outputs in the model (Xue et al. 2015) |
| Natural oasis ecosystem | <50, 50-61.4, >61.4 | Water demand for ensuring natural ecosystem (million $m^3$) | Calculated outputs in the model (Xue et al. 2015) |
| Spring irrigation | Sufficient, insufficient | Water demand accounting for 35% of total consumption in spring | Results of stakeholder interviews |
| Crop yields | <235.9, 235.9-239.7, >239.7 | Crop yields (thousand tons) | |
| Market price | High, low | Crop market price | Results of stakeholder interviews |
| Salt-removing system | Good, poor | Salt-removing engineering | Results of stakeholder interviews |
| Water for man-made shelterbelt | <12989.7, 12989.7-27168, >27168 | Water demand for man-made shelterbelt growth ( thousand $m^3$) | Qira water resources planning report (2013) |
| Water-saving efficiency | <0.43, 0.43-0.62, >0.62 | Water-saving efficiency in the irrigation system | Qira water resources planning report (2013) |
| Available agricultural water | <0.1268, 0.1268-0.1518, >0.1518 | Agricultural water supply (billion $m^3$) | Qira water resources planning report (2013) |
| Public water-saving awareness | <50%, 50%-80%, >80% | Percent of farmer surveys | Survey results |
| Irrigation regime | Drip irrigation, sprinkler irrigation, flood irrigation | Three irrigation regime | Hotan Water Resources Planning (2013) |
| Agricultural irrigation quantity | <98520.7, 98520.7-100625.1, >100625.1 | Agricultural irrigation (thousand $m^3$) | Qira water resources planning report (2013) |

| | | | |
|---|---|---|---|
| Industrial water demand | Under 270.4, over 270.4 | Water demand for industrial development (thousand m$^3$) | Qira water resources planning report (2013) |
| Water for people and animals | Under 2307.8, over 2307.8 | Water demand for people and animals (thousand m$^3$) | Qira water resources planning report (2013) |
| Water for urban greenbelt | Under 80, over 80 | Water demand for urban greenbelt (thousand m$^3$) | Qira water resources planning report (2013) |
| Agricultural income | <0.30, 0.30-0.35, >0.35 | Agricultural total income (billion RMB) | Statistical Yearbooks of Xinjiang Province (2002–2013) |
| Biodiversity | Good, medium, poor, extremely poor | Biodiversity based on species and growth | Results of stakeholder interviews |
| Groundwater safety | High, medium, low, extremely low | Groundwater condition based on depth and quality | Results of stakeholder interviews |
| Drinking-water security | <8.6, 8.6-25.6, 25.6-44.2, >44.2 | Drinking-water people with risk (thousand people) | Qira water resources planning report (2013) |
| Soil salinization | <10.08, 10.08-16.80, 16.80-21, >21 | Area insulted from salinization (ha) | Qira water resources planning report (2013) |
| Grassland degradation | Good, medium, poor, extremely poor | grassland growth condition | Results of stakeholder interviews |
| Land desertification | <104.26, 104.26-259.77, 259.77-628.5, >628.5 | Land area suffered from desertification disaster (km$^2$) | Results of stakeholder interviews |

---

## Referee Comment (RC3) · N. Rivers-Moore (Referee) · 6 Apr 2017

Specific comments:

- Referencing and citations generally adequate; certain references need a suffix of a or b: Poppenorg et al.; Liu et al.

- Spelling of certain references or citations, and consistency between these: Siew and Döll; Duespohl et al. Chen & Pollino (as opposed to Chen et al. p5 line20); more efficient citing of multi-year papers by the same authors: Egoh et al. 2007-2008; Xue et al. 2016 a, b (p4, line 11).

[Figure]

- Lynan 2006 or 2007?

- Burgess and Chilvers (2006) not referenced

- Charnley vs. Chamley?

- Aims and methods written in current tense, rather than past tense

- I have reviewed the MS acknowledging that the MS was written by non-first language English scientists, and tried to separate content/ concepts from style. The MS would benefit from editing by a first-language English editing service (although the MS is generally well written).

- Key Bayesian network texts e.g Jensen and Nielsen; Kjaerulff and Madsen – appear to have been omitted.

General comments:

- The study has been undertaken within a relatively small catchment. It was not clear from the text how many stakeholders there were, what the population density and size is, and how stakeholders where identified.

- Large portions of Sections 2.2 and 3 would seem to fit better in to an introduction.

- It was not clear where the methods end, and the results and discussion begin. This needs to be clearer.

- Using the methods provided from the text, it would be difficult to replicate this study. The process of identifying nodes and nodes states is not defined, and nor was the calculation of parent node probabilities or population of conditional probabilities. This is a critical issue- at the minimum, these tables should be included as a data appendix.

- Furthermore, the Bayesian Network appears to be overly complex, such that the population of the CPTs would have also been a complex procedure. There is no indication that there has been model output verification (although admittedly this is often a failing

of BN papers).

- The research appears to be fairly sound in terms of stakeholder participation and model sensitivity analyses, but weak in terms of BN development process, and data use. While useful within a broader IWRM perspective, the actual BN approach is not particularly innovative, and seems like another replication of the approach used by Cain (2001).

In summary, I do not believe that this research is innovative enough to warrant publication, or explained in sufficient detail to allow for replication. My recommendation would be to reject.
* * *

---

## Author Comment (AC3) · 8 Apr 2017

**Response to Reviewer #3:**

**Title:** *Development of a participatory Bayesian network model for integrating ecosystem services into catchment-scale water resources management*
**Authors:** *Jie Xue, Dongwei Gui, Jiaqiang Lei, Fanjiang Zeng, Rong Huang, Donglei Mao*
**Manuscript ID:** *hess-2016-618*

**The authors would like to thank you for the time you invested in reviewing this manuscript. We would also like to thank you for your insightful comments on this revised version of the manuscript.**

**According to your comments, we have carefully modified the manuscript. The corresponding revised manuscript with the modifications shown in the document is attached following the responses to the comments.**

Specific comments:
- Referencing and citations generally adequate; certain references need a suffix of a or b: Poppenorg et al.; Liu et al.
**Response: Thank you for highlighting the mistake of references in our manuscript. We have added the corresponding suffix in the text and references as suggested.**

- Spelling of certain references or citations, and consistency between these: Siew and Döll; Duespohl et al. Chen & Pollino (as opposed to Chen et al. p5 line20); more efficient citing of multi-year papers by the same authors: Egoh et al. 2007-2008; Xue et al. 2016 a, b (p4, line 11).
**Response: Thank you for point this out. We have changed them in the text and references as suggested.**

- Lynan 2006 or 2007?
**Response: Thank you for point this out. It should be Lynam et al. (2007). We have changed it in the text and references.**

- Burgess and Chilvers (2006) not referenced
**Response: Thank you for point this out. We have added it in references.**

- Charnley vs. Chamley?
**Response: Thank you for point this out. It should be Charnley and Engelbert (2005). We have checked it in the text and references correctly.**

- Aims and methods written in current tense, rather than past tense

**Response: Thank you for highlighting this problem. To improve the grammatical errors of our manuscript, we have invited a professor in the field of water resources, who is a proficient English speaker, to go through our manuscript. Please see the changed part in the manuscript.**

- I have reviewed the MS acknowledging that the MS was written by non-first language English scientists, and tried to separate content/ concepts from style. The MS would benefit from editing by a first-language English editing service (although the MS is generally well written).

**Response: Thank you for point this out. With respect to the language problem of our paper, we have sought out a scientific language service (http://lucidpapers.com/) to polish my manuscript (Please see the certificate). Furthermore, we have invited a professor of our institution, who is a proficient English speaker, to go through our manuscript.**

**Certificate** | **LucidPapers** Editing for authors and publishers

| | |
|---|---|
| **Reference number**: 2016-102502 | **Date**: 08 November 2016 |
| **Contact author**: Jie Xue | **Manuscript**: Development of a participatory Bayesian network model for integrating ecosystem services into catchment-scale water resources management |

- Key Bayesian network texts e.g Jensen and Nielsen; Kjaerulff and Madsen – appear to have been omitted.

**Response: Thank you for highlighting this problem. We have added the two references (Nielsen and Jensen, 2009, and Kjaerulff and Madsen, 2008) in the text and references as suggested.**

**Cited references are listed as follows:**

**Nielsen, T. D., Jensen, F. V.: Bayesian networks and decision graphs. Springer Science & Business Media, 2009.**

**Kjaerulff, U. B., Madsen, A. L.: Bayesian networks and influence diagrams. Springer Science Business Media, 2008.**

General comments:

- The study has been undertaken within a relatively small catchment. It was not clear from the text how many stakeholders there were, what the population density and size is, and how stakeholders where identified.

**Response: Thank you for your comment. While the study area (The Qira oasis area, Northwest China) is a relatively small catchment, it is able to effectively reflect and deal with the actual water management issues due to the handleability in participatory modelling. Table 1 in the manuscript has listed the number of the stakeholders (12 people). According to the characteristics of water-related ecosystem services and functions, the stakeholders with six sectors are identified and determined by water supply and demand features under the discussion and negotiation of research team. According to Burguess and Chilvers (2006), the number of stakeholders should be kept as small as possible, and also able to completely represent their own viewpoints. Therefore, the head and professional of water sectors are selected as representatives by their more comprehensive understanding, rather than by the population density and size. To describe it more clearly, we have improved the statement. Please see changed the sentences in Section 4.1.**

**Cited references are listed as follows:**

**Burgess, J., and Chilvers, J.: Upping the ante: a conceptual framework for designing and evaluating participatory technology assessments. Sci. Public Policy, 33, 713-728, 2006.**

- Large portions of Sections 2.2 and 3 would seem to fit better in to an introduction.

**Response: Thank you for your suggestion. We agree that part of Sections 2.2 and 3 can be put in the introduction. Considering the balance of manuscript structure, the introduction will become tedious, if large portions of Sections 2.2 and 3 are put into the introduction. Therefore, we think that the introduction in the manuscript only presents the question of research, methods and framework associated with Sections 2.2 and 3. The detailed methods and framework should be put in the Sections 2.2 and 3. According to your suggestion, we have improved part of descriptions in the manuscript.**

- It was not clear where the methods end, and the results and discussion begin. This needs to be clearer.

**Response: Thank you for your suggestion. To set the structure more clearly, we have adjusted subtitles in the manuscript as suggested. Please see the changed subtitles in the manuscript.**

- Using the methods provided from the text, it would be difficult to replicate this study. The process of identifying nodes and nodes states is not defined, and nor was the calculation of parent node probabilities or population of conditional probabilities. This is a critical issue- at the minimum, these tables should be included as a data appendix.

**Response: Thank you for your comments. As you said, we have not provided additional information to detail the variables, states, and data information. To explain more clearly, we have added a supplementary file as Appendix A to provide detailed information about**

variables, states, and data information. This Appendix A provides a summary of the variables, states, and information sources used to elicit the CPTs. Moreover, the sentence "The states representing each variable and information used to elicit the CPTs are explained in Appendix A" is inserted in the manuscript. Please see the added Appendix A in the revised manuscript with the modifications following the responses to the comments.

- Furthermore, the Bayesian Network appears to be overly complex, such that the population of the CPTs would have also been a complex procedure. There is no indication that there has been model output verification (although admittedly this is often a failing of BN papers).

**Response: Thank you for your comment. Since water resources management is a very complex interdisciplinary issue, this implies that the water management issue is inevitably a complicated process. The advantage of BNs can integrate various factors into the structure of BN models for sustainable water management. We agree that the BN model developed display a complex structure, but it also provide an effective tool for decision-making and management in integrated water resources management framework. Due to the nonrepeatability of many variables particularly output variables, the model output verification is based on sensitivity analysis (such as Poppenborg and Koellner, 2014), expert assessment (e.g., Zorrilla et al., 2010), or combination analysis between the two. Section 3.2 in revised version has explained the model output verification using combination analysis between the sensitivity analysis and expert assessment. The expert assessment is based on the evaluation methods of Zorrilla et al., 2010. To describe it more clearly, we have improved the statement in Section 3.2.**

**Cited references are listed as follows:**

**Poppenborg, P., and Koellner T.: A Bayesian network approach to model farmers' crop choice using socio-psychological measurements of expected benefits of ecosystem services, Environ. Modell. Softw., 57, 227–234, 2014.**

**Zorrilla, P., Carmona G., Hera Á. D. L., Varela-Ortega C., Mart ńez-Santos P., Bromley J., and Henriksen H. J.: Evaluation of bayesian networks in participatory water resources management, upper guadiana basin, spain, Ecol. Soc., 15, 634-634, 2010.**

- The research appears to be fairly sound in terms of stakeholder participation and model sensitivity analyses, but weak in terms of BN development process, and data use. While useful within a broader IWRM perspective, the actual BN approach is not particularly innovative, and seems like another replication of the approach used by Cain (2001).

**Response: Thank you for your comment. We agree that the BN development is very complex process, and the participatory procedure of stakeholders is also a very time-, energy- and money-consuming process. However, the aim of IWRM is to seek a comprehensive, holistic and interdisciplinary way to effectively manage the water resources (Savenije and Zaag,**

2008). Therefore, the BN model is developed to deal with such complex process involved in multiple factors. Recently, many BN models (such as Wang et al., 2009, Chan et al, (2010), Chen and Pollino et al. (2012), and Mamitimin et al. (2015)) are developed to solve the IWRM issues in such complex process. We believe that Cain (2001) is the same as Bromley (2005), Marcot et al. (2006), Kragt et al. (2009), and Pollino and Henderson (2010) to provide extensive and detailed guidelines in the participatory process (including BN development process and data preparation). Due to flexible and open system tool, the BN development is to deal with specific water management problems associated with IWRW framework. Instead of replicating the approach such as Cain (2001), our BN model developed is to integrate ecosystem services into IWRM framework. In the participatory process, we only used the theory of those guidelines (such as Cain (2001)) to develop the feasible BN model structure. With respect to innovation or novelty of our manuscript, we have explained it in the final comment.

Cited references are listed as follows:

Savenije, H. H. G., and Zaag P. V. D.: Integrated water resources management: concepts and issues, Phys. Chem. Earth Parts A/b/c, 33, 290-297, 2008.

Wang, Q. J., Robertson D. E., and Haines C. L.: A Bayesian network approach to knowledge integration and representation of farm irrigation: 1. model development. Water Resour. Res., 45, 142-143, 2009.

Chan, T., Ross H., Hoverman S., and Powell B.: Participatory development of a bayesian network model for catchment-based water resource management, Water Resour. Res., 46, 759-768, 2010.

Chen, S., and Pollino C.: Good practice in Bayesian network modelling, Environ. Modell. Softw., 37, 134–145, 2012.

Mamitimin, Y., Feike T., and Doluschitz R.: Bayesian network modeling to improve water pricing practices in northwest china, Water, 7, 5617-5637, 2015.

Cain, J. D.: Planning Improvements in Natural Resources Management. Guidelines for Using Bayesian Networks to Support the Planning and Management of Development Programmes in the Water Sector and Beyond, 124pp., Centre for Ecology and Hydrology: Wallingford, UK, 2001.

Bromley, J.: Guidelines for the use of Bayesian networks as a participatory tool for Water Resource Management, Centre for Ecology and Hydrology: Wallingford, UK, 2005.

Kragt, M. E.: A beginners guide to Bayesian network modelling for integrated catchment management. Landscape Logic technical report no.9., Landscape Logic, Australia, viewed 11 December 2012, 2009.

Pollino, C. A., and Henderson C.: Bayesian networks: A guide for their application in natural resource management and policy, Integrated Catchment Assessment and Management Centre, Fenner School of Environment and Society. Australian National

**University, Canberra, 2010.**

In summary, I do not believe that this research is innovative enough to warrant publication, or explained in sufficient detail to allow for replication. My recommendation would be to reject.

**Response: Thank you very much for your time and your pertinent comments to our manuscript. While you didn't approve of our contribution in the innovation, we believe that our manuscript is innovative enough to deal with an important issue, integrating ecosystem services into IMRW framework. With respect to the innovation or novelty of our manuscript, our explanation is as follows:**

**(1) Managing water ultimately seeks benefits obtainable from water allocation to maximize human wellbeing provided by ecosystem services, which are defined as a wide range of goods and services provided by ecosystems for human welfare (Millennium Ecosystem Assessment, 2005).** **There is an increasing consensus on the importance of integrating ecosystem services into integrated water resource management. However, IWRM and ecosystem services have evolved into closely similar concepts, and face challenges linked to the coupling between them in terms of conceptualization and implementation (Cook and Spray (2012)). According to Jewitt, 2002, the main problem at this juncture is that IWRM does not consider ecosystems as "users" of water in allocation. Faced with such problem, we proposed an ecosystem services–based IWRM framework to build a bridge between the ecosystem services and IMRW from conceptualization and implementation. Thus, the proposed ecosystem services–based IWRM framework can successfully couple the concepts and implementations between ecosystem services and IWRM for achieving sustainable water resource management.**

**(2) Management and decisions of IWRM need decision support system tools and involve the participation of stakeholders. Stakeholder involvement will provide effective coordination among various conflicts in the decision-making process, transparently and practically (Cain, 2001; Bromley, 2005; Kragt, 2009; Zorrilla et al., 2010). Moreover, Bayesian networks (BNs), which are graphical decision support system tools allowing "what-if" analysis through probability inference can effectively deal with the complexity and uncertainty involved in specific environmental modelling problems (Bromley, 2005; Pollino and Henderson, 2010; Liu et al., 2013).** **Although stakeholder engagement in the decision process exchanges viewpoints to share new knowledge and solutions to common issues, few attempts have been made to confirm whether BNs developed by active stakeholder involvement and negotiation can assist and achieve common consensus to integrate ecosystem services into IWRM. This paper develops a participatory Bayesian network model to perform a proposed ecosystem services-based water management framework under public participation. Thus, the participatory Bayesian network developed can effectively integrate ecosystem services into IWRM framework for transdisciplinary and sustainable**

**water management.**

We would like to highlight the innovation or novelty of our manuscript briefly as:

- **Proposing a framework which integrates ecosystem services into IWRM system.**
- **Developing a participatory Bayesian network model to perform the ecosystem services-based water management framework proposed under public participation.**
- **Participatory Bayesian network model effectively provides the support of transdisciplinary water management, achieving the aim of integrating ecosystem services into IWRM framework.**

Moreover, we believe that our work is not a replication or simple application. The BN model developed provides an open and transparent system to support integrating ecosystem services into IWRM framework. More importantly, the structure uncertainty of BN model caused by poor knowledge and understanding can be timely updated by new knowledge and data available. The developed BN model can be appropriate for the areas in which there is intense competition for water between human activities and ecosystems, particularly in arid regions worldwide.

We insist that our research is innovative enough to give adequate reason in solving the relevant problems. Should you have any questions or need further information from us, please let us know. Thanks again.

Cited references are listed as follows:

[revised manuscript text omitted]